# Functional genomics of chitin degradation by *Vibrio parahaemolyticus* reveals finely integrated metabolic contributions to support environmental fitness

Landon J. Getz[ID][1][☉], Oriana S. Robinson[1][☉], Nikhil A. Thomas[ID][1,2]*

1 Department of Microbiology and Immunology, Dalhousie University, Halifax, Nova Scotia, Canada,
2 Department of Medicine (Infectious Diseases), Dalhousie University, Halifax, Nova Scotia, Canada

☉ These authors contributed equally to this work.
* n.thomas@dal.ca

**Data availability statement:** All data is available without restriction. All data are in the manuscript and/or Supporting Information files.

**Funding:** This work was funded by the Natural Sciences and Engineering Research Council (NSERC) of Canada with a Discovery Grant (RGPIN 05807) awarded to NT. LG is the recipient of a Canadian Vanier Graduate Scholarship

## Abstract

*Vibrio* species are marine prokaryotes that inhabit diverse ecological niches, colonizing abiotic and biotic surfaces. These bacteria are vital players in the global carbon cycle, assimilating billions of tonnes of chitin for carbon (and nitrogen) metabolites. Many bacterial proteins involved in the process-including chitinases, sugar transporters, and modifying enzymes-have been well studied. However, the genetic functional interplay and key drivers of *Vibrio* competitive survival in the presence of chitin as the dominant carbon source is not understood. To address this question, we carried out transposon sequencing (Tn-seq) to determine the genetic fitness of *Vibrio parahaemolyticus* mutants grown on chitin as a sole carbon source. Along with validating known *Vibrio* genes associated with chitin metabolism, our data newly identified vital roles for an unclassified OprD-like import chitoporin and a HexR family transcriptional regulator. Furthermore, we functionally implicated HexR in regulating multiple physiological processes involved in *V. parahaemolyticus* environmental survival including carbon assimilation and cell growth, biofilm formation, and cell motility. Under nutrient limiting conditions, our data revealed a requirement for HexR in filamentous cell morphology, a critical trait for *V. parahaemolyticus* environmental fitness. Therefore, a vital import porin and genomic regulation mediated by HexR support multiple physiological processes for *Vibrio* chitinolytic growth and environmental fitness.

## Author summary

*Vibrio* species are key contributors to ocean carbon cycling by degrading and assimilating robust chitin polymers. These waterborne and seafood associated bacteria also infect humans leading to considerable morbidity and mortality. To better understand the *Vibrio* aquatic environment to infection transmission cycle, we have conducted an original and innovative population-based transposon sequencing (Tn-seq) study to identify *V. parahaemolyticus* genes that contribute to competitive fitness under exclusive chitinolytic growth. We discovered and functionally linked an unclassified single channel

and Killam Doctoral Award. The funders had no role in study design, data collection and analysis, decision to publish, or preparation of the manuscript.

**Competing interests:** The authors have declared that no competing interests exist.

β-barrel outer membrane porin to chitooligosaccharide translocation. This OprD-like porin is structurally distinct from the prototype trimeric ChiP porins of *Vibrio* species thus revealing an additional mechanism for large N-acetylglucosamine polymer transport. Additionally, we discovered that the HexR transcriptional regulator is a key factor that supports *V. parahaemolyticus* environmental survival. Under nutrient limiting conditions HexR contributes towards regulated biofilm formation, cell motility, and gene regulation of central carbohydrate metabolism pathways. Interestingly, our data reveals that under nutrient limiting conditions HexR is important for *V. parahaemolyticus* cell filamentation, a key trait linked to *Vibrionaceae* environmental survival. Taken together, our Tn-seq approach has enhanced the knowledge of *Vibrio*-chitin interactions, with new discoveries that inform on an integrated genomic network that supports *Vibrio* sp. environmental fitness.

## Introduction

*Vibrio* species are marine bacteria that colonize biotic and abiotic surfaces in their environmental niches. These bacteria can exist free-living, but can also attach to marine phytoplankton, copepods, mollusks, and fish. Water and seafood borne human infections mainly by pathogenic *Vibrio cholerae*, *Vibrio parahaemolyticus*, and *Vibrio vulnificus* account for significant global morbidity and numerous deaths [1,2], especially during regional outbreaks and following natural disasters. Furthermore, some *Vibrio* species are major operational and safety threats to seafood and aquaculture industries [3]. Therefore, advanced knowledge of *Vibrio* species in the environment and risks associated to human health and seafood safety are of primary concern.

The *Vibrionaceae* are moderate halophiles with considerable metabolic versatility. Unlike most bacteria, *Vibrio* species can degrade robust chitin polymers found within phytoplankton and animal exoskeletons, extracting nitrogen and carbon to support growth [4]. As chitin is the 2nd most abundant carbon polymer on Earth (after cellulose), its degradation is paramount for environmental carbon cycling [5]. Critically, members of *Vibrionaceae* generate chitinases *de novo* and secrete these enzymes to effectively degrade chitin into smaller N-acetyl glucosamine (GlcNAc) oligosaccharides which are eventually imported and processed for cytoplasmic carbon assimilation and essential catabolic activities (discussed in detail below). The importance of environmental chitin in *Vibrio* biology is functionally integrated into various physiological processes. Specifically, *Vibrionaceae* respond to chitin by chemotactic sensing behaviour, expressing surface pili, forming adherent biofilms, upregulating DNA uptake machinery (competence), and altering cell morphology [6–10]. Accordingly, the *Vibrionaceae* have evolved complex genetic and biochemical adaptations to effectively locate and utilize chitin within their environmental niches.

The biochemistry of *Vibrio*-mediated chitin degradation has been widely studied [4,11–14]. Initially, secreted chitinases act to degrade chitin polymers into smaller oligosaccharides. Bacterial outer membrane ChiP porin proteins form highly selective trimeric channels that facilitate transport of the chitin oligosaccharides into the periplasm where chitodextrinases and β-N-acetylhexosaminidases act to cleave these polymers into mono- or disaccharides. These smaller sugars are then moved into the cytoplasm by phosphotransferase systems and ATP binding cassette (ABC) family of transporters [4,15]. Following cytoplasmic transport and phosphorylation of the GlcNAc monomer, GlcNAc-6-P is converted to GlcN-6-P by a deacetylase (NagA), followed by conversion to fructose-6-P and $NH_3$ by a deaminase (NagB). Fructose-6-P feeds into the parallel pathways for general metabolism, glycolysis and the

pentose-phosphate pathway [4]. Moreover, the periplasmic chitin binding protein (CBP) binds to a conserved cytoplasmic membrane ChiS transcriptional regulator which controls the expression of various *Vibrionaceae* genes involved in chitinolytic activities, thus providing a sensing link to support regulated chitin degradation [16,17]. Additionally, quorum sensing regulators contribute to linking chitin degradation to natural competence, catabolite repression, and cellular filamentation, physiological features that are critical environmental fitness determinants [6,7,10,18–21].

Microarray expression and mutant strain studies have functionally implicated various genes and regulatory factors in *Vibrio*-mediated degradation of chitin [13]. Moreover, a genomic *V. cholerae* study has identified many genes that are key contributors for host infection and/or dissemination into the aquatic environment [22]. Nonetheless, concepts pertaining to coordinated genomic regulation remain modestly understood. More specifically, genes that are functionally important to *Vibrio* fitness (competitive survival) under exclusive chitinolytic conditions (sole carbon source) has not been evaluated at an integrated genomic level. To address this and other related questions, we have conducted a population-based transposon sequencing (Tn-seq) study to identify *V. parahaemolyticus* genes that contribute to competitive fitness and survival under exclusive chitinolytic growth. Overall, the data revealed a finely integrated metabolic network that supported environmental fitness relating to *Vibrio* chitinolytic growth. Our data newly implicates an unclassified porin protein in chitooligosaccharide import and the HexR transcriptional regulator as key factors that support *V. parahaemolyticus* chitinolytic survival. Additional experiments revealed significant HexR dependence for biofilm formation, motility, and gene regulation of central carbohydrate metabolism pathways. Lastly, we find that under nutrient limiting conditions, HexR is required for *V. parahaemolyticus* filamentous cell morphology, a key trait linked to *Vibrionaceae* environmental survival.

## Results

### Identification of *V. parahaemolyticus* biologically essential genes for glycolytic or chitinolytic growth

We developed an efficient genetic transposition system to initially identify biologically essential genes within *V. parahaemolyticus* RIMD2210633. The bacterial strain was selected for stable streptomycin resistance and its genome sequenced to provide a baseline reference (see methods and S1 Data). Next, we used an efficient biparental conjugation approach to introduce a suicide plasmid containing a mariner-type transposon into *V. parahaemolyticus*. Cells undergoing transposition were selected on minimal medium with glucose (see methods) supplemented with chloramphenicol and streptomycin to select against donor bacteria.

Next, isolated transposon mutant populations were then grown in liquid minimal media containing either glucose or colloidal chitin as their sole carbon source and incubated until mid-log growth. Genomic DNA was isolated from the population of transposon mutants and served as the reference data set of established (viable) mutants with transposon insertions. The TRANSIT insertion sequencing analysis software was used to determine essential genes in the glucose condition (minimal medium + glucose) as well as the chitin condition (minimal medium + 0.4% colloidal chitin) and compare the two conditions to find conditionally essential genes [23] (S1 Data). This was done by first using a hidden markov model (HMM) approach to identify essential regions of the genome in each condition, followed by a zero-inflated negative binomial (ZINB) approach to compare the two conditions. First, each condition was subjected to an HMM analysis which locally determines differences in transposon insertion density and provides essentiality calls for genes: growth advantage, non-essential,

growth disadvantage, and essential. HMM analysis of the glucose condition determined 9 genes whose mutations conferred a growth advantage, 3671 genes that were non-essential, 223 genes whose mutations generated a growth defect, and 663 genes that were essential for growth (Fig 1A and 1B). In the chitin condition, mutations in 82 genes conferred a growth advantage, 3725 genes were non-essential, mutations in 148 genes conferred a growth disadvantage, and 611 genes were labelled essential for growth (Fig 1A and 1B). 3547 genes were non-essential and 539 were essential in both conditions.

### Tn-seq identifies fitness determinants linked to chitinolytic growth and validates known *Vibrio* genes in chitin metabolism

To assess if the chitin selection media was performing as expected, we compared our data to modeled *Vibrio* chitin catabolic pathways [4,14] with an expectation to identify a subset of genes previously linked to chitin metabolism. As expected, conditionally essential genes for the chitin condition included *nagA* (*vpa0038*), *nagB* (*vp0829*), *chiS* (*vp2478*), *cbp*

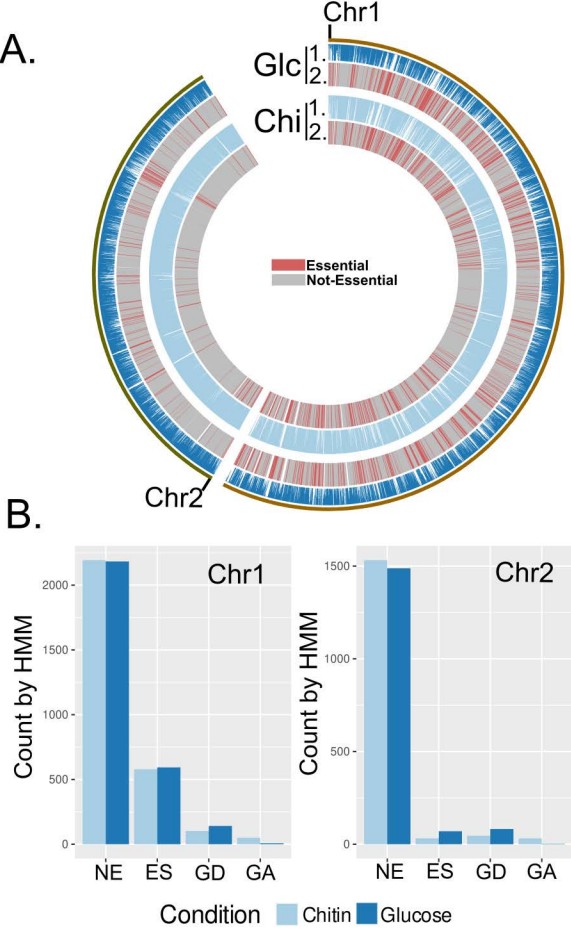

**Fig 1. Summary of Tn-seq analysis. (A)** Circos plot showing transposon insertion data across both chromosome 1 and chromosome 2; (1.) the loci indicated as essential (combined growth disadvantage and essential from HMM analysis) and non-essential (combined growth advantage and non-essential from HMM analysis) (2.). Red loci are essential, while grey loci are non-essential. Both glucose (Glc and Chitin (Chi) growth condition datasets are shown. **(B)** Genes indicated as non-essential (NE), essential (ES), growth defect (GD), or growth advantage (GA) by HMM analysis for both chromosomes.

(*vp2479*), almost the entire *chb* operon (*vp2479-vp2488*, except *vp2485*), one of three encoded β-N-acetylhexosaminidases (*vp2486*), and a chitodextrinase (*vp0832*) (Fig 2A, 2B, and 2C). Importantly, these genes comprise the two major chitin catabolism pathways in *Vibrio* spp. and all had significant p-values by ZINB (S1 and S2 Data). Furthermore, most (but not all) of the type II secretion system (T2SS) associated genes were identified as essential (Fig 2D), consistent with the established role of this system in secreting chitinase enzymes that degrade chitin polymers. These data suggest that our applied Tn-seq approach for *V. parahaemolyticus* was successfully interrogating the biological system related to chitin metabolism. Critically, the data identifies each sequential enzymatic reaction required for intracellular oligomeric GlcNAc catabolism (Fig 3). Interestingly, *nagE* (*vp0831*), the glucosamine kinase *(vp2485)*, the cellobiose PTS (*vp2635-vp2637*), ChiP (a chitoporin, *vp0760*) (Fig 2E), two of the three encoded β-N-acetylhexosaminidases (*vp0755;* p=0.283 and *vp0545;* p=0.407), as well as all the encoded chitinases (*vp2338*, *vpa0055*, *vpa1177*, *vp0619*) were not conditionally essential in the chitin condition (Fig 3). Some of these observations can potentially be explained by known functional genetic redundancy within the genome (e.g., multiple chitinase and hexosaminidase genes) or bifunctional enzymatic activities for some proteins. In contrast, the conditional non-essential finding for the chitoporin (Vp0760) raised the perplexing question as to how large chitooligosaccharides, as a sole carbon source, would enter the bacterial cell to support growth. Interestingly, an unclassified OprD-like porin, encoded by *vp0802*, was identified as essential for chitinolytic growth (Fig 2F), a finding that is novel since these porins typically transport amino acids and other carboxylic acid compounds, and not N-acetylated compounds like chitooligosaccharides (see detailed section below). Furthermore, the striking fitness defect for *vp1236* (p=0.000016) (S2 Data) revealed a novel discovery for a helix-turn-helix transcriptional regulator with respect to specific chitinolytic growth (see detailed sections below).

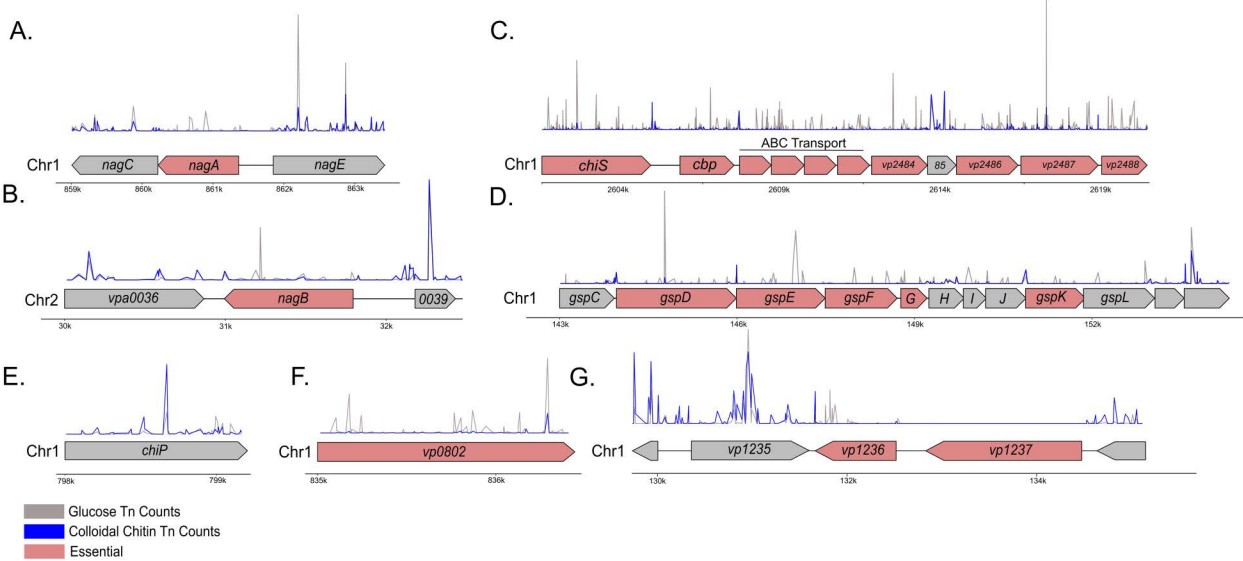

**Fig 2. Tn-seq insertion data indicating specific gene regions.** (**A**) *nagE-nagAC* operon, (**B**) *nagB* and surrounding genomic region, (**C**) *chiS* and the *chb* operon, (**D**) *gsp* operon for T2SS, (**E**) *chiP*, (**F**) *vp0802*. (**G**) *vp1236* and surrounding genomic region. Chromosome numbers are indicated with approximate kilobase markers for location. The gene annotation for each region is shown, with genes demonstrating statistically underrepresented transposon insertions by ZINB analysis shown in red. Insertion counts for each genomic region are shown as peaks for both the chitin and glucose growth conditions. *vp1237* is an essential gene (glutamate decarboxylase) under both growth conditions.

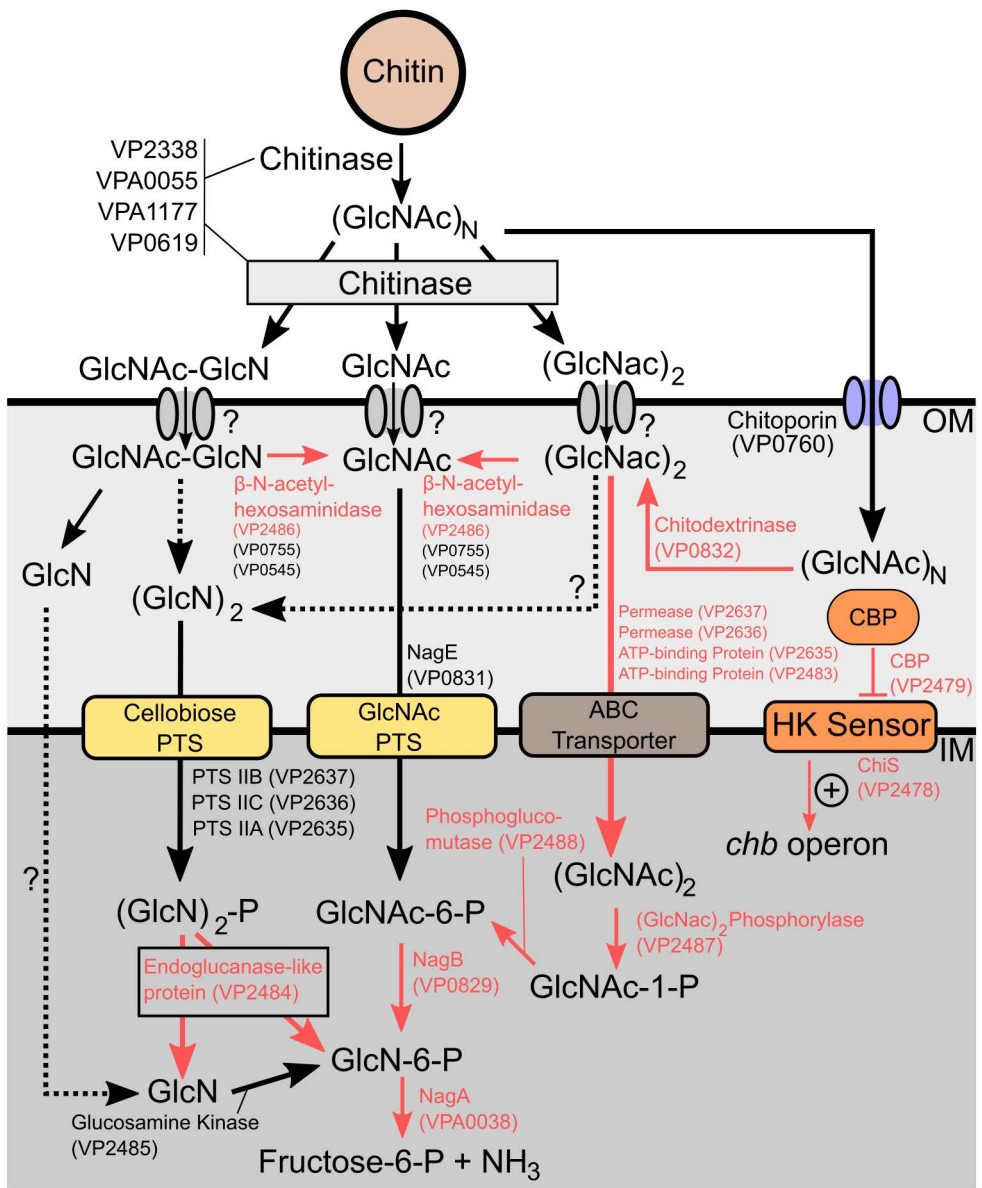

**Fig 3. Genes essential for chitin metabolism as identified by Tn-seq.** A pathway diagram for catabolism of chitin to fructose-6-phosphate is shown, with component enzymes and their associated locus tags indicated. Chemical conversions which are known to occur, but do not have associated enzymes are shown with a question mark and dotted arrow. Conversions and locus tags shown in red are indicated as conditionally essential (p < 0.05) for growth on chitin by ZINB analysis of the Tn-seq dataset.

## The *V. parahaemolyticus* Type II Secretion System (T2SS) is required to support efficient chitinolytic growth

T2SS are multicomponent protein assemblies that secrete proteins into the extracellular environment [24]. *Vibrio* species are known to secrete their cognate chitinases via the T2SS [25]. Strikingly, while none of the chitinase genes were essential on their own (S1 Data), many genes identified in the *gspCDEFGHIJKL* type II secretion system operon were essential for growth on chitin (Fig 2D). While some of the genes in this operon did not meet the

p-value cutoff of 0.05, many approached significance and were essential by HMM analysis (S2 Data). *gspD*, *gspE*, *gspF*, *gspG*, and *gspK* were statistically underrepresented in the chitin growth condition, while *gspC* and *gspI* approached significance. *gspH*, *gspJ*, and *gspL* were not underrepresented in the chitin growth condition (Fig 2D). Importantly, *gspH*, *gspJ*, and *gspI* encode minor pseudopilin subunits, and this data would suggest that mutations in these genes do not affect the function of the T2SS with respect to supporting growth on colloidal chitin. Taken together, these data suggest that a functional T2SS is a critical fitness determinant for *V. parahaemolyticus* chitinolytic growth.

## An unclassified OprD-like porin is required for efficient growth on colloidal chitin

*Vibrio* species are known to utilize a trimeric ChiP porin complex to selectively import chitooligosaccharides into the periplasm (Figs 4A and S1) [26–29]. The porin complex is in the bacterial outer membrane where it supports facilitated diffusion of chitooligosaccharides driven by binding affinities within the hydrophilic porin channels [27,28,30]. ChiP is often considered essential for growth on chitin as this chitoporin is found in most *Vibrio* species [13]. In contrast, a study in *V. parahaemolyticus* RIMD2210633 (the same pandemic strain in this study) found that ChiP was not essential for growth on chitin flakes [31]. This latter observation is concordant with our Tn-seq data for ChiP being non-essential for chitin uptake. Instead, our Tn-seq data revealed a significant growth defect for mutants with insertions within *vp0802* (p<0.0001, Fig 2F), an open reading frame that putatively encodes a protein with approximately 35-40% sequence similarity to OprD-like single channel porins in various Proteobacteria [32,33] and is broadly conserved in various *Vibrio* DNA sequence database entries (S1 Fig). This was unexpected, especially since OprD family porins are considered selective to substrates with carboxylic acid (-COOH) groups, including amino acids and some antibiotics (e.g., carbapenems) [34]. AlphaFold2 structural modelling analyses [35] of Vp0802 predicts a single channel beta-barrel presumably located in the outer membrane of bacterial cells, with remarkable structural similarity to OprD family porins [34] (Figs 4A and S1). Moreover, SignalP analysis identified a high confidence putative signal peptide sequence, consistent with protein export beyond the cytoplasmic membrane, presumably with an outer membrane localization for the barrel structure (S2 Fig).

To investigate the contribution of Vp0802 relating to chitinolytic growth, we constructed a null mutant strain deleted for the *vp0802* DNA coding sequence (Δ*vp0802*) for comparison to wild type *V. parahaemolyticus* (RIMD2210633, parent strain). After an initial lag phase of approximately 24 hours, wild type bacteria exhibited efficient logarithmic growth and entered a stationary phase at 36 hours (Fig 4B) (S3 Data, for statistical analyses between specific time points for a given strain). Viable cell counts (achieved by plating on LBS agar) appeared to decline however there was not a decrease in the colloidal cell culture density or any visible aggregation. This may represent a temporary state of a viable but not culturable condition (see discussion) as a sporadic increase in viable bacteria at 66 hours was observed. When compared to the wild type, the Δ*vp0802* strain had a significant growth defect when colloidal chitin was the sole carbon source and exhibited an extended lag phase until 66 hours (Fig 4B). After 66 hours, the Δ*vp0802* strain did eventually exhibit growth as revealed by an increase in cell number between 66 and 72 hours post inoculation. Genetic complementation with a plasmid encoding *vp0802* restored chitinolytic growth, albeit delayed compared to the parental strain (S3 Fig). These data suggest that *vp0802* contributes to efficient chitinolytic growth. Furthermore, bacteria deficient for *vp0802* expression can utilize an alternate and temporally separate mechanism to produce chitinolytic growth.

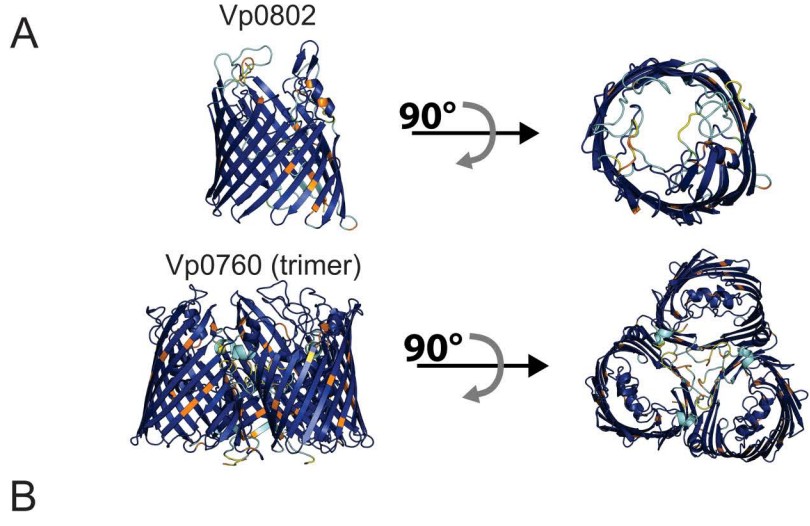

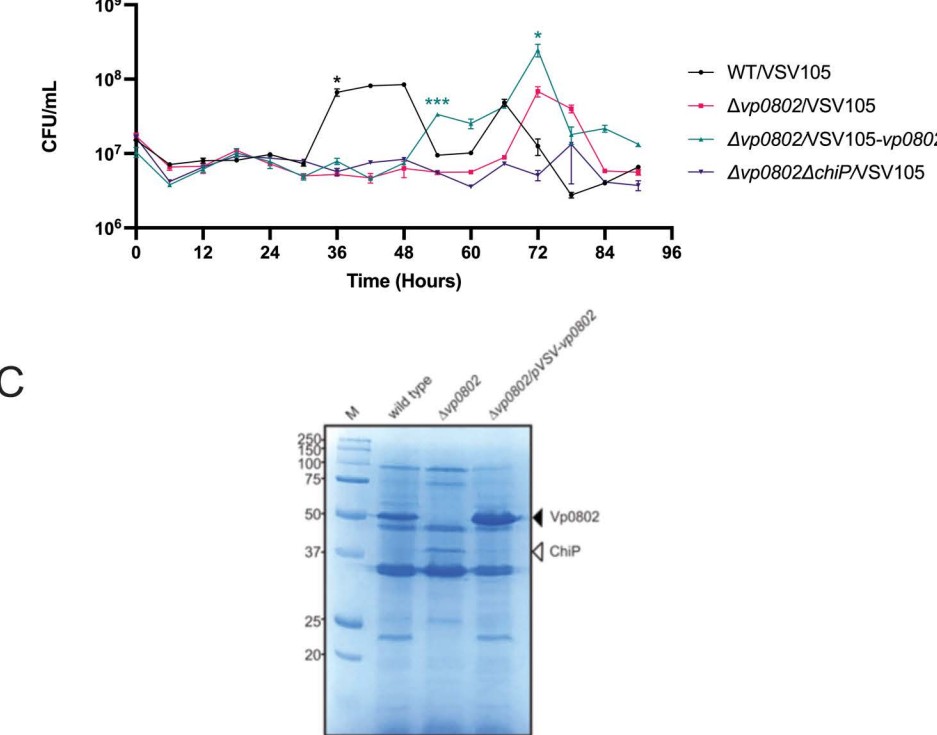

**Fig 4. Vp0802 is an outer membrane OprD-like β-barrel porin that supports efficient chitinolytic growth. (A)** Alphafold2 structural prediction of Vp0802 compared with ChiP trimer, coloured by predicted local distance difference test (pLDDT) confidence values. **(B)** Chitinolytic growth of indicated bacterial strains as determined by viable plate counts. Note that wild type cells with high growth (t=48 to 54 hours) enter a putative viable but not culturable (VBNC) condition after depleting the media of colloidal chitin. Statistical significance was determined with an unpaired t-test between wild type and each of the strains at the indicated time points (n=3, *:p < 0.05). **(C)** Coomassie stain of bacterial outer membrane fractions isolated from the indicated strains at early stationary phase of growth for the respective strains. The black arrowhead points to Vp0802 and the white arrowhead points to ChiP with mass spectrometry validation of the protein species. M, molecular mass standards.

Next, we set out to determine the cellular localization of Vp0802. After 80 hours of colloidal chitin growth, bacterial cells were harvested, followed by sonication and lauryl sarcosine sulfate treatment to enrich an outer membrane fraction [26,36] for SDS-PAGE analyses. A dominant protein species was visible at approximately 50 kDa for the parental strain, consistent with the predicted molecular mass of 49.25 kDa for processed (mature) Vp0802 (Fig 4C). Notably, this protein species was absent from the outer membrane preparation derived from the Δvp0802 strain that exhibited delayed growth and was clearly restored in the genetically complemented strain (Fig 4C). Interestingly, a dominant protein species at approximately 40 kDa was apparent in the Δvp0802 strain but missing in the parental and genetically complemented strains (Fig 4C). We hypothesized that this apparent 40 kDa protein species was the ChiP chitoporin (Vp0760), as its predicted mature molecular mass is 38.33 kDa. Therefore, we subjected the respective 50 and 40 kDa protein species to mass spectrometry analyses to determine their identities. As expected, the 50 kDa protein outer membrane protein species was determined to be Vp0802 by peptide matching (S4 Fig). Furthermore, the 40 kDa protein species was conclusively identified as ChiP. Moreover, we generated a vp0802, chiP double mutant strain which was subsequently demonstrated to be severely compromised for growth

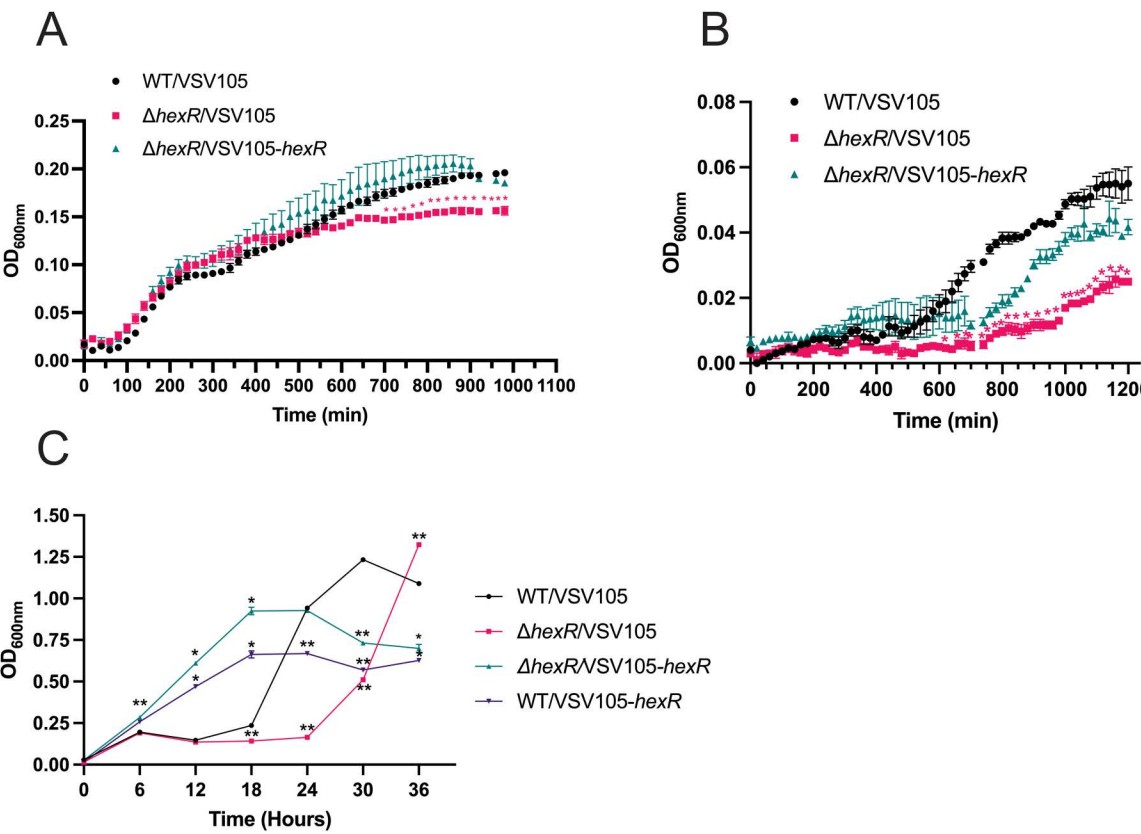

**Fig 5. The role of HexR in supporting *V. parahaemolyticus* growth under different nutrient conditions. (A)** Growth of indicated strains in LB medium. Strains were cultured overnight in LBS and normalized to starting OD600nm of 0.025. Cell density readings were taken every 20 minutes for 14 hours at 37°C in a VictorX5 Multi Label plate reader. **(B)** Growth of indicated strains in MM9 media supplemented with glycerol. **(C)** Growth of indicated strains in LB Spent Media. Strains were inoculated into spent media following growth in LBS at a starting $OD_{600nm}$ = 0.025. Optical density ($OD_{600nm}$) measurements were taken in duplicate for each strain every 6 hours for 36 hours. Statistical significance was calculated at the indicated time points using an unpaired t-test against wild type (n=3, *: p<0.05, **: p<0.01).

on colloidal chitin (Fig 4B). Genetic complementation of the *vp0802, chiP* double mutant with a plasmid expressing Vp0802 or ChiP partially restored chitinolytic growth (S3 Fig), suggesting that these two chitoporins are the main outer membrane uptake mechanisms for large chitooligosaccharides. Collectively these data indicate that Vp0802, encoding an OprD-like porin, is localized to the outer membrane of *V. parahaemolyticus* cells grown in colloidal chitin as a sole source and that ChiP outer membrane protein levels are minimal in that context. In bacterial cells without Vp0802, the ChiP chitoporin was expressed and accumulated in the outer membrane to support chitinolytic growth.

## Vp1236 is an important regulator for growth on diverse nutrient sources

The genomic location of Vp1236 exhibits conserved gene synteny in various *Vibrio* species (S5 Fig). These proteins are believed to be linked to central carbon metabolism based on sequence similarity to the HexR/MurR/RpiR family of transcriptional regulators. Notably, Vp1236 exhibits 59% amino acid identity (77% similarity) with *Pseudomonas aeruginosa* (PAO1) HexR where its role in carbohydrate utilization was initially characterized [37]. We opted to use HexR nomenclature for Vp1236 from here on, based on HexR being linked to six carbon sugar metabolism in other bacteria (such as glucose and other hexoses) and other known roles in central carbon metabolism [38–40]. Vp1236 shares 95% sequence identity with Vc1148 of *V. cholerae* and neither of these proteins have been functionally implicated in chitinolytic growth [13,31]. To investigate the contribution of HexR in *V. parahaemolyticus* fitness, a chromosomal *hexR* deletion (*ΔhexR*) was generated and compared to the wild type strain for growth in both rich and minimal nutrient sources. Under static conditions in rich LB media, wild type bacterial growth produced a sigmoidal growth curve, entering an initial stationary phase at 300 minutes and then showed a steady rate of 'secondary' growth that reached a plateau around 800 minutes with a maximum $OD_{600nm}$ 0.196 (Fig 5A). The *ΔhexR* strain produced a similar initial sigmoidal growth trend to wild type bacteria but its secondary growth rate began to exhibit a statistically significant reduction in growth with a maximum $OD_{600nm}$ of 0.157. Genetic complementation with a plasmid expressing HexR (pVSV105-*hexR*) restored the growth pattern to that of wild type bacteria (Fig 5A). These data suggest that the *ΔhexR* strain growth defect was due to the functional absence of HexR.

In the case of nutrient limiting Marine M9 media (MM9) containing glycerol as the sole carbon source, the *hexR* mutant growth defect was further exacerbated (Fig 5B). Glycerol is metabolized into dihydroxyacetone-phosphate (DHAP) which can enter either the glycolytic or gluconeogenic pathways [41]. While the wild type *V. parahaemolyticus* strain did have an extended lag phase compared to rich (LB) conditions, the sigmoidal growth trend remained the same with a maximum $OD_{600nm}$ of 0.054. In contrast, *ΔhexR* exhibited an extended lag phase in the first 750 minutes followed by statistically significant reduction in growth where cultures reached a maximum $OD_{600nm}$ of 0.025 (Fig 5B). Genetic complementation of *ΔhexR* restored the growth phenotype albeit delayed, back to the pattern observed for the wild type strain.

Entrance into stationary phase prompts significant metabolic changes and shifts the flux of carbon [42]. Likewise, adaptations of bacteria into stationary phase or nutrient depleted conditions include significant changes to cell morphology and physiology to allow the cell to persist during environmental stressors (e.g., changes in pH or nutrient depletion) [43]. To further investigate the growth of *ΔhexR* in nutrient depleted conditions, we conducted cell growth experiments using filtered LB spent media derived from 18 hour wild type cultures (shaking, aerobic, late stationary phase). Under aerobic shaking spent media conditions, wild type bacteria displayed a long lag phase of approximately 16 to 18 hours before entering logarithmic growth. The *ΔhexR* strain lag phase was even longer at 24 hours, further suggesting that

these bacteria have a defect in nutrient assimilation. Strikingly, the *ΔhexR* genetically complemented strain did not exhibit a lag phase and grew faster than wild type bacteria between 6 to 18 hours. After 18 hours, HexR overexpression appeared to limit additional growth as cell density reached a plateau (between 18-24 hours) and declined after 24 hours (Fig 5C). In contrast, between 18-30 hours wild type bacteria were still growing as revealed by a continued increase in cell density. Moreover, *ΔhexR* bacteria exhibited a delayed increase in cell density between 24-36 hours. This suggested that *hexR* expression provides an initial growth advantage in nutrient limited environments, but its dysregulated expression or absence is inhibitory in extremely low nutrient conditions. Indeed, wild type *V. parahaemolyticus* overexpressing HexR did not exhibit a lag phase and had an initial increased growth rate over the normal wild type strain (Fig 5C). Furthermore, the growth trends of the overexpression HexR strain and the *ΔhexR* genetically complemented were similar and both strains entered stationary phase at 18 hours. Taken together, this suggests that while the overexpression of HexR provides an initial growth advantage in nutrient-limited environments, ultimately wild type bacteria with inherent coordinated central carbon metabolism mediated by HexR exhibit efficient growth.

## HexR is an important regulator for growth on chitin as a sole carbon source

To validate the Tn-seq *hexR* observations relating to chitinolytic growth, we performed growth assays using MM9 supplemented with 0.4% colloidal chitin as the sole carbon source. Unexpectedly, while the *ΔhexR* strain had a statistically significant growth defect in the first 16 hours, it consistently outgrew the wild type strain by 20 hours and continued this faster rate of growth until reaching stationary phase (Fig 6A). The wild type strain showed slower growth kinetics and reached a similar final cell density by 54 hours. Notably, when *ΔhexR* was genetically complemented, the strain returned to wild type pattern of growth (Fig 6A). Therefore, the *ΔhexR* strain displayed dysregulated and an altered growth rate on chitin as a sole carbon source.

The observation of faster dysregulated chitinolytic growth for the *ΔhexR* strain was paradoxical to the Tn-seq gene fitness data where a gene fitness defect was observed for *hexR* transposants (insertional mutants). A key difference in the experiments is that Tn-seq is a mixed population-based competitive growth context, whereas *ΔhexR* bacteria grown in isolation (i.e., pure culture) represents an isogenic condition. Therefore, to investigate the competitive fitness of wild type and *ΔhexR* bacteria, we generated a fluorescently tagged wild type strain with a transposon system capable of expressing stable green fluorescent protein (GFP +). A similar approach was performed to generate three *ΔhexR* transposant strains that were non-fluorescent (GFP-) (see methods). This allowed for an experimental setup with a tractable mixed culture competition assay. Critically, we initially confirmed that all the transposant strains were unchanged for LB growth compared to their parental strains (original sources) (S6 Fig). Furthermore, we mapped the chromosomal transposon insertion site for each strain and confirmed that the disrupted genes do not contribute to chitinolytic genetic fitness based on our Tn-seq dataset (S6 Fig and S1 Data). Next, following co-inoculation of the transformant strains at the same starting $OD_{600nm}$ into MM9 (0.4% colloidal chitin), serial dilutions of the cultures were plated over a time course of 48 hours. There was no significant change in growth between wild type or a *hexR* mutant transposant strain during the first 24 hours post-inoculation, however, after 48 hours the wild type transposant (GFP+) had significantly outgrown *ΔhexR* bacteria (GFP-)(Fig 6B). The average competition index (CI) derived from three independent competition experiments between wild type (GFP+) and each of the three *hexR* transposants (GFP-) at 48 hours was 0.4 (Fig 6C). Incorporating GFP expression in the *hexR* mutant (GFP marker swap) also demonstrated a competitive growth defect (S6 Fig). Given that a CI < 1 indicates a fitness defect, these results support the interpretation that cells

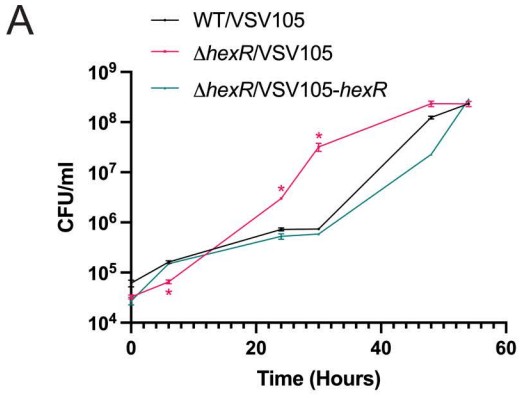

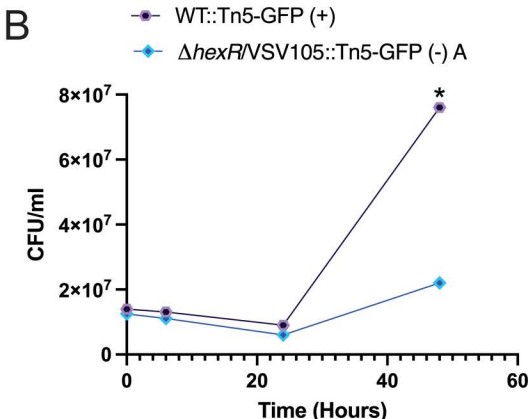

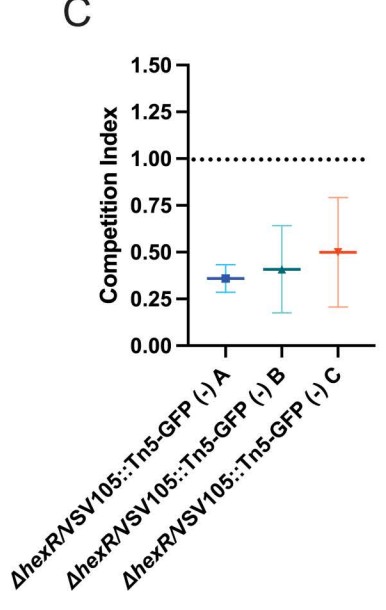

**Fig 6. HexR is a fitness determinant for *V. parahaemolyticus* on chitin as a sole carbon source. (A)** Overnight LBS cultures of strains were inoculated into M9 Minimal Media containing colloidal chitin as a sole carbon source. Samples were collected every 6-12 hours for 54 hours. Serial dilutions were plated to determine the number of cells present. Statistical significance was determined with an unpaired t-test between wild type and the indicated strain at indicated time points (n=3, *:p < 0.05). **(B)** Competition assay to assess fitness defect of Δ*hexR* compared to wild type *V. parahaemolyticus*. Overnight LBS cultures of wild type GFP (+) transposant and Δ*hexR* GFP (-) transposant (strain

A) were inoculated together into MM9 (0.4% colloidal chitin) at a starting OD600nm of 0.025. Samples were collected every 6-12 hours for 48 hours. Serial dilutions were plated to determine the number of fluorescent cells and total cells present. Statistical significance was measured with an unpaired t-test (n=3, *: p<0.05). **(C)** Competition Indices (CI) of three independent GFP (-) *hexR* mutant transposant strains (A, B, C). The transposon insertion site for each strain was determined and the respective strains did not differ for growth when compared to the Δ*hexR* parent (S6 Fig). The individual GFP (-) *hexR* mutant transposants were subjected to a head-to-head competition assay with the WT GFP (+) strain in MM9 (0.4% Colloidal Chitin). The CI for three experiments (n=3) were plotted, CI < 1 indicates a fitness defect.

lacking functional *hexR* were at a significant competitive disadvantage in mixed culture chitinolytic growth conditions. Importantly, this competition assay data validates the observations from the Tn-seq data and implicate contextual *hexR* expression as a critical fitness determinant, especially under nutrient limiting conditions.

## HexR is a regulator for biofilm formation and cell differentiation under nutrient limiting conditions

The data implicating HexR as a fitness determinant in nutrient limiting conditions suggests that HexR functionality would contribute to physiological factors that are important for *V. parahaemolyticus* environmental persistence. Notably, biofilms contribute significantly to the aquatic survival of *V. parahaemolyticus* and require considerable investment in the synthesis and export of polysaccharides [44]. Biofilm formation is additionally important for efficient chitinolytic growth which occurs post-cellular attachment to chitin polymers. As Δ*hexR* bacteria displayed growth defects under nutrient limiting conditions, this led us to investigate its biofilm phenotype. As expected, in MM9 media (with glycerol and casamino acids) wild type *V. parahaemolyticus* displayed biofilm formation. In contrast, Δ*hexR* exhibited a statistically significant reduction in biofilm formation even when cell number was normalized (Fig 7). Genetic complementation restored biofilm formation to wild type levels indicating that HexR contributes to proper biofilm formation under nutrient limited conditions. Critically,

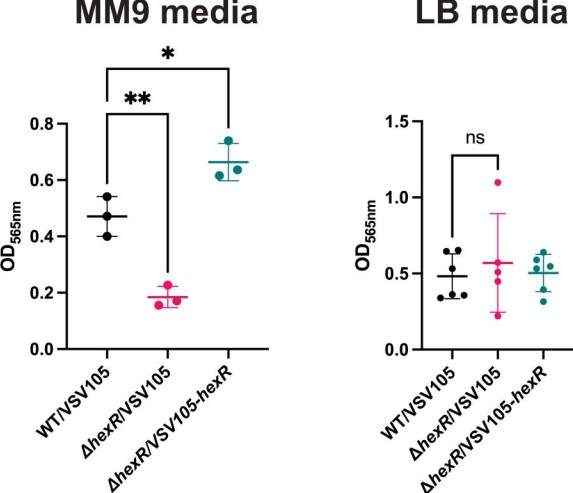

**Fig 7. HexR contributes to *V. parahaemolyticus* biofilm formation under nutrient limiting conditions.** Biofilm formation of strains grown in MM9 (0.4% Glycerol, 0.2% Casamino Acids) or LB for 18 hours at 37°C. Each dot represents an individual sample. Statistical significance was calculated using one-way ANOVA (n=9, *:p < 0.05, **: **p** < 0.01).

we tested all the strains for biofilm formation in rich LB media and observed no difference, suggesting that the role of HexR in biofilm formation is likely nutrient or context dependent (Fig 7).

Our data implicating HexR for biofilm and chitin association prompted us to evaluate cell shape and cell differentiation as previous studies have reported that dysregulated central carbon metabolism often leads to defects in cell morphology [45, 46]. Moreover, for growth on chitinaceous sources, *V. parahaemolyticus* is known to exhibit elongated cell morphologies as a response to environmental cues [21]. Observations of all strains cultured in LB medium revealed no apparent differences in cell shape, with slightly curved rods being uniformly apparent (Fig 8A). In contrast, wild type bacteria cultured in nutrient limiting MM9 (glycerol) or MM9 with colloidal chitin developed a filamentous subpopulation of cells. Strikingly, Δ*hexR* cells remained homogenously and strictly curved rod-shaped and were devoid of filamentous cells (Fig 8A). Genetic complementation of Δ*hexR* restored the filamentous cell subpopulation. This phenotype was extremely robust and was quantified across multiple fields of view in microscopy observations. Across 50 fields of view (FOV) with a minimum of 30 cells, the filamentous cells made up approximately 20-32% of the total cells observed in both wild type and genetically complemented strain backgrounds but were completely absent for the Δ*hexR* strain (Fig 8B). Under the MM9 glycerol condition, we measured cell lengths from the respective cultures and observed a significant reduction in cell length for the Δ*hexR* cells compared to wild type bacteria (Fig 8C). When the same strains were cultured in MM9 supplemented with colloidal chitin, the Δ*hexR* cells were predominantly observed to be short rods in contrast to the wild type which exhibited a mix of filamented cells and short rods (Fig 8D). Based on these observations, we conclude that HexR contributes to *V. parahaemolyticus* cell filamentation under nutrient limiting conditions.

## HexR activity is important for regulated swimming motility but not surface swarming

*V. parahaemolyticus* cells are highly motile in the aquatic environment where the different cell morphologies (swimming and swarming cells) utilize different flagellar systems to promote dissemination and colonization [47]. Therefore, we initially set out to assess motility by stab inoculating each strain in soft agar allowing for quantifiable motility measurements via radial growth distance. As expected, measurements of radial growth confirmed wild type *V. parahaemolyticus* motility (Fig 9A and 9B). In contrast, *hexR* mutants had a statistically significant increase in radial growth indicative of increased cell motility. This increase in swimming motility was lost following genetic complementation with *hexR*. This data reveals that HexR impacts on regulated motility, and more specifically that its absence leads to increased motility.

*V. parahaemolyticus* surface swarming is associated with cell elongation and expression of lateral flagella in a *lafK* gene dependent manner [21,48]. Cells typically exhibit swarming when cultured using rich (complex) nutrients, low iron conditions, and in the presence of calcium [48]. We observed equivalent swarming (no significant difference) for both wild type and Δ*hexR* bacteria with characteristic growth 'flares' at the periphery of swarmer colony growth (Fig 9C and 9D). It should be noted that there was considerable variability in swarming radius for the 6 technical replicates for each strain ranging from 45-80 millimetres, therefore subtle differences could be masked. Nonetheless, in each case, the localized flare bacteria were expressing *lafK* as demonstrated by the strains harboring a transcriptional fusion to the *lafK* promoter (P*lafK-gfp*), thus producing green fluorescent cells (Fig 9E). Notably, Δ*hexR* bacteria were elongated similar to wild type cells under these nutrient replete conditions. For all strains

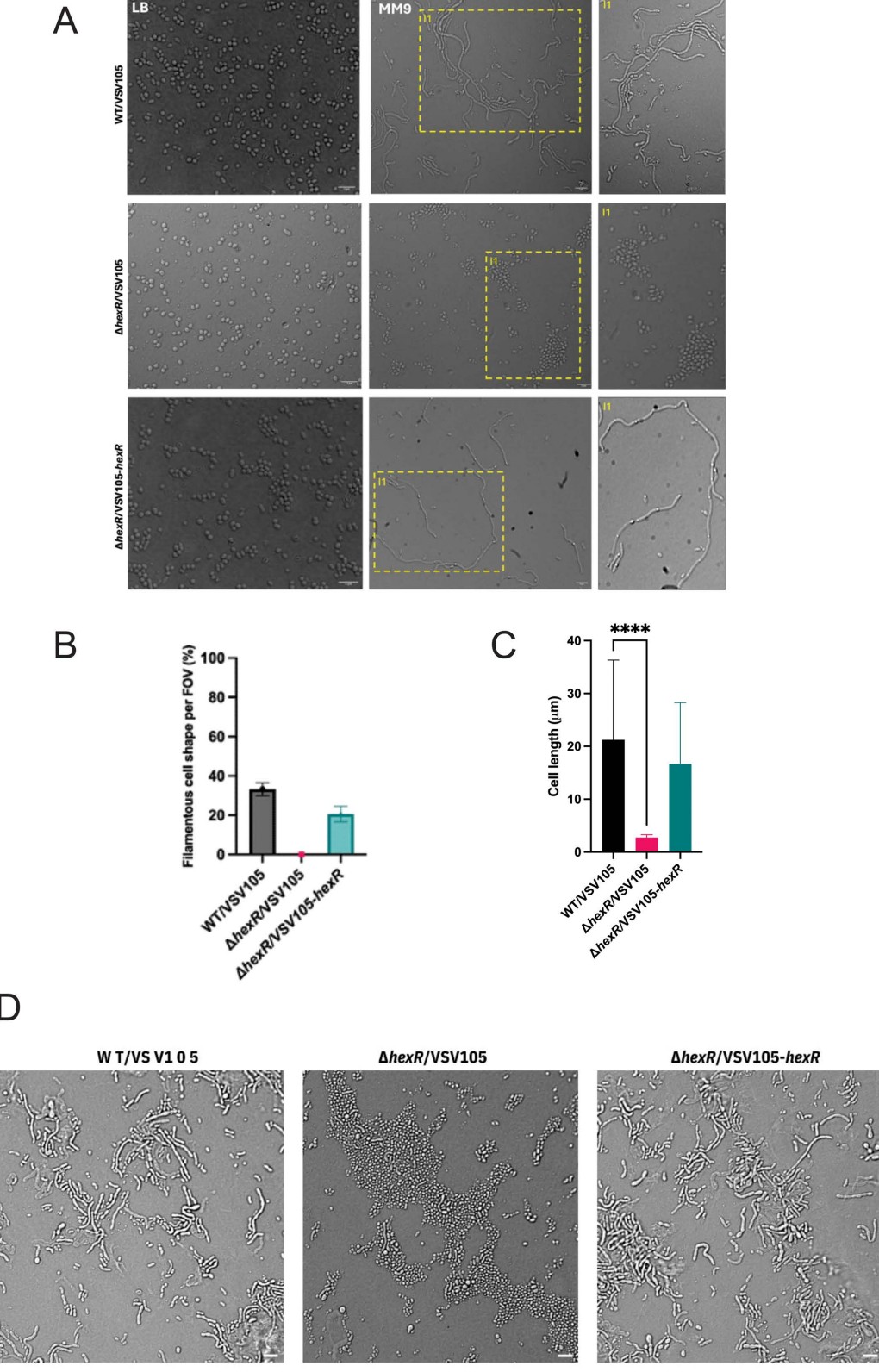

**Fig 8. HexR contributes to cell filamentation in nutrient limiting conditions. (A)** Representative microscopy of wild type (WT), *hexR* mutant, and *hexR* complement strains (top to bottom) in LB and MM9 (0.4% glycerol). Cells were observed at 100x. Scale bar = 5μm. Insets (I1) identify cell morphologies for each genotype. **(B)** Quantification of

filamentous cell subpopulation for the indicated strains. Cells were imaged with microscopy after 16 hours of growth in MM9 (0.4% Glycerol). All cells were counted in a field of view (FOV), and filamentous cells further enumerated to determine the % of filamentous cells per FOV. **(C)** Measurement of cell length of for bacteria cultured in MM9 (0.4% glycerol) conditions. A minimum of 50 cells across 5 independent observations were measured using Image J. Data were plotted with standard deviation and statistical significance was calculated using an unpaired t-test (n=50, ****: p<0.0001). Mean wild type cell length was 21μm, mean *hexR* mutant cell length was 3μm, and *hexR* complement cell length was 17μm. **(D)** Representative microscopy of wild type, *hexR* mutant, and *hexR* complement strains cultured in MM9 (0.4% colloidal chitin). Cells were observed at 100x. Scale bar = 5μm.

tested, we were unable to stimulate a swarming phenotype under nutrient limiting conditions of growth (e.g., minimal or colloidal chitin solid media) for unknown reasons. Nonetheless, under nutrient rich conditions Δ*hexR* bacteria can produce swarmer cells based on elongated cell morphology, *lafK* promoter activity, and a swarming phenotype on the agar surface.

## HexR regulates central carbon metabolism-associated genes important for cell fitness

HexR is a helix-turn-helix (HTH) global transcriptional regulator of central carbon metabolism (CCM) genes in many Proteobacteria [38, 39]. In the case of *V. parahaemolyticus*, bioinformatic analyses predict HexR to bind at a conserved pseudo-palindromic motif (TGTAATTAAATTACA) within promoter regions of various genes [49](S4 Data). Depending on the genomic context, HexR may act as a negative or positive regulator of gene expression [39]. HexR binding downstream of a promoter is thought to repress gene expression, whereas HexR binding upstream of a promoter is viewed to positively regulate gene expression. Therefore, we hypothesized that disruptions to CCM via the chromosomal deletion of *hexR* would likely contribute to gene expression differences with an outcome that impacts genetic fitness. To investigate this, we assessed gene promoter activities of two predicted *V. parahaemolyticus* HexR regulon members: Glucose-6-P isomerase (*pgi*), and Glycogen Debranching Enzyme (*glgX*), in both wild type and Δ*hexR* strains. These putative HexR regulon members were selected based on their respective high score for HexR consensus binding sequences identified by RegPrecise [49] and their linkage to central carbon metabolism Proteobacteria [39]. Real-time quantitative cell-based luciferase reporter assays were performed to measure promoter activities.

In the cases of *pgi*, Δ*hexR* bacteria demonstrated significantly higher promoter activity than wild type bacteria at the 4 and 4.5 hour time points over a time course experiment (Fig 10A). Genetic complementation using a plasmid expressing HexR repressed *pgi* promoter activity below that of wild type levels, suggestive of HexR being a negative regulator of *pgi* expression. Consistent with this view, a putative HexR binding site (TGAAAAAAAATTACA) was localized downstream of the predicted *pgi* promoter element (S7 Fig).

Comparison of *glgX* promoter activity between wild type and Δ*hexR* bacteria revealed significantly reduced transcriptional activity for Δ*hexR* bacteria (Fig 10B). Genetic complementation with a plasmid expressing HexR resulted in *glgX* promoter activity at levels above that observed for wild type bacteria. These data suggest that *glgX* expression is positively regulated by HexR, a finding consistent with a putative HexR binding site (AGTAATTAAATTACA) being identified upstream of a the predicted *glgX* promoter (S7 Fig).

Collectively, the promoter activity data for the HexR regulon members *pgi*, and *glgX*, suggest that HexR is involved in regulating central carbon metabolism associated genes. This data further suggests a pivotal role for HexR mediated gene regulation in chitinolytic growth, as derived N-acetylglucosamine (GlcNAc) import and processing requires coordinately

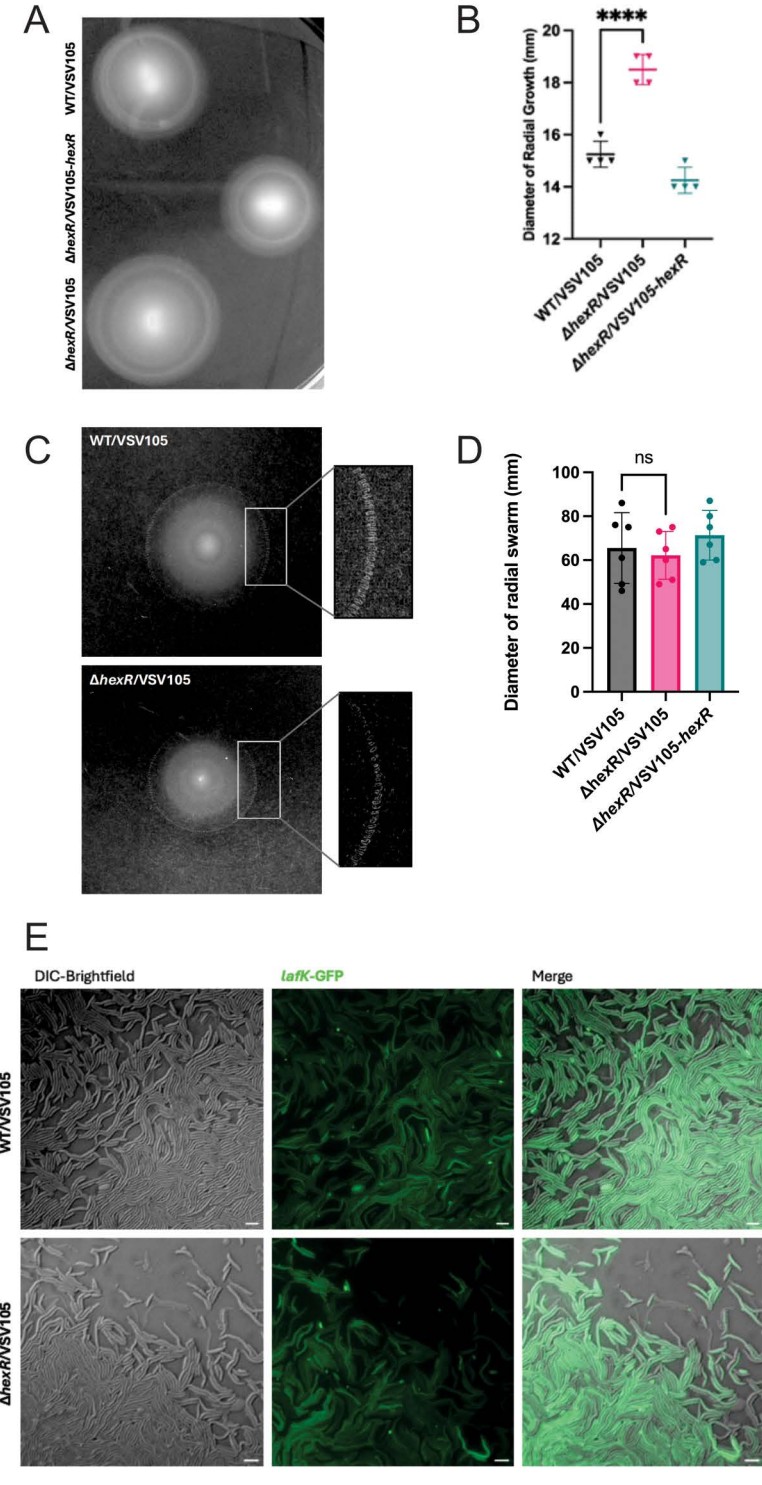

**Fig 9. HexR contributes to regulated swimming motility but not swarming. (A)** Evaluation of swimming motility for the indicated strains using motility agar. The bacterial strains were cultured overnight in LBS, normalized to OD600 of 0.05 in 1x MM9 salts and stab inoculated into 0.2% (w/v) agar LBS plates. **(B)** Quantification of swimming phenotype by measuring the diameter of radial growth. Measurements were recorded after 5 hour incubation at 37°C. Statistical significance was calculated using one-way ANOVA (n=4, ****:p < 0.0001). Each dot represents a biological replicate. **(C)** Swarmer cell differentiation for the indicated strains was induced (see methods) and 1μL of cultures were spotted onto fresh BHI agar (4mM CaCl$_2$, 50μM 2,2-Bipyridil). Plates were sealed and incubated at 24°C overnight. Swarmer cell flares were imaged using CannoScan 5600F (Cannon) scanner. Images were processed on ImageJ.

**Table 1.** (Continued)

(D) Quantification of swarming phenotype by measuring the diameter of radial growth. Measurements were recorded after overnight growth. Statistical significance was calculated using one-way ANOVA (n=5). Each dot represents a biological replicate. (E) Visualization of swarmer cells obtained directly from the peripheral flare region of growth. Peripheral cells from wild type and *hexR* mutant strains each expressing a P*lafK*-GFP reporter were stamped onto 1% agarose pads and imaged with DIC and GFP fluorescence on Z2 Axio Imager (Zeiss). All samples were viewed at 63x. Scale bar 5μm.

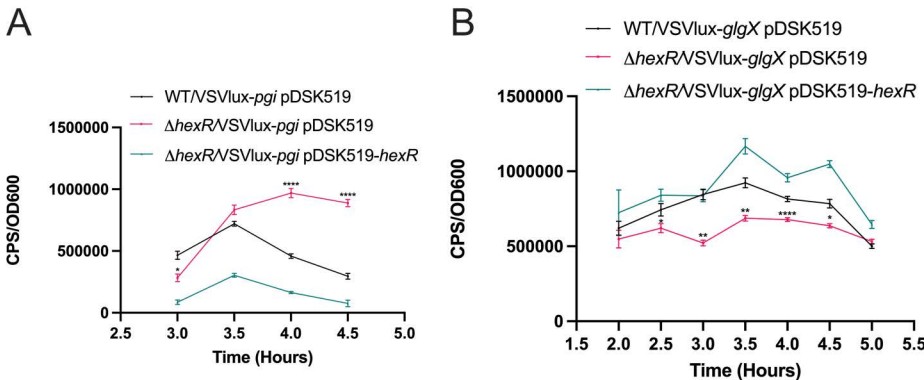

**Fig 10. HexR directly regulates *pgi* and *glgX* expression in *V. parahaemolyticus*.** (A) Luciferase Reporter assay of *pgi* promoter region in wild type (WT), ΔhexR, and *hexR* genetically complemented strain. (B) Luciferase reporter assay of *glgX* promoter region in WT, ΔhexR and a *hexR* genetically complemented strain. The strains were grown overnight in LBS and inoculated at a starting OD600nm of 0.05 into LB. CPS (photon counts per second) and OD600 readings were recorded at the indicated time points. Complementation of *hexR* was performed using plasmid pDSK519-*hexR*. Statistical significance was calculated with an unpaired t-test compared to wild type (n=3, *: $p<0.04$, **: $p<0.01$, ****: $p<0.0001$).

regulated enzymatic activities (Glucose-6-P isomerase, glycogen debranching enzyme) for efficient carbon assimilation leading to essential amino acid and nucleotide biogenesis.

## Discussion

This study reports the use of Tn-seq to functionally interrogate genes relating to chitin degradation and has newly implicated two broadly conserved *Vibrio* genes in environmental fitness. Firstly, we discovered that HexR activity is functionally relevant for *V. parahaemolyticus* with HexR significantly impacting processes of chitinolytic growth, biofilm formation, cell motility, and cell filamentation. Notably, these processes are all known to be key aspects for *Vibrio* environmental survival and persistence [10,19,50–52]. Secondly, this study identified an outer membrane porin protein as a new *Vibrio* spp. chitoporin, structurally distinct from the known trimeric ChiP chitoporin of *Vibrio* species [27–29,53]. The discovered chitoporin is present in many *Vibrio* species, and likely forms a single channel OprD-like porin structure (see Fig 4A). In the case of *V. parahaemolyticus*, our data suggests that this porin is a functionally relevant chitooligosaccharide import mechanism for these bacteria. Lastly, this study validates and expands on chitinolytic genetic associations known for *V. cholerae* within a separate pandemic isolate of *V. parahaemolyticus* O3:K6.

Chitinolytic growth and metabolism in *Vibrio* species has been extensively studied however most approaches have been reductionist and performed in isogenic contexts. The use of Tn-seq to study broad collective gene contributions to a specific selective pressure is powerful on many levels. Notably, Tn-seq experiments allow for millions of population level fitness

impacts on a genome wide scale. Moreover, specific and more intricate gene fitness observations can be discerned due to generational growth parameters during the experiment. A previous gene expression microarray profiling study in *V. cholerae* was foundational in identifying and characterizing genes involved in the *Vibrio* chitinolytic cascade, however, defined RNA sampling time points were used making gene fitness determinations difficult to establish. An extensive Tn-seq study in *V. cholerae* was conducted to evaluate gene fitness in context of dissemination from within an animal model host to pond water (in addition to other growth conditions) [22]. Interestingly, the study did identify *V. cholerae* HexR (Vc1148) as a gene fitness determinant in pond water, which might be considered a limited nutrient environment depending on its content and source. The exact functionality of *V. cholerae* HexR deficiency (cell phenotypes) was not evaluated in the pond water context, however, our Tn-seq data for *V. parahaemolyticus* HexR in defined chitinolytic growth conditions is concordant with those gene fitness findings. Furthermore, our study has revealed specific HexR dependent phenotypes that are critical for environmental survival, including biofilm formation, cell motility, and regulation of CCM genes. Moreover, the dependency for HexR in *V. parahaemolyticus* filamentous cell differentiation under nutrient limiting conditions is a key discovery. Cell filamentation has been identified as a crucial chitin surface attachment mechanism for both *V. cholerae* and *V. parahaemolyticus* biofilm cell populations [10,21], although the mechanisms governing the transition from curved rods to filamentous cells is not understood. The role of HexR in cell filamentation is therefore an initial starting point to further investigate this important *Vibrionaceae* feature.

Another major advance from this study is the discovery of an outer membrane porin that is conditionally required to confer efficient growth to *V. parahaemolyticus* when chitin is the sole carbon source. A previous study reported a deletion of the *chiP* gene from *V. parahaemolyticus* still resulted in bacterial growth on chitin flakes [31]. This was difficult to rationalize since it contradicts the widely held view that trimeric ChiP is the main chitoporin for all *Vibrio* species. Our Tn-seq data for *chiP* (along with generating a Δ*chiP* strain) further validated the non-essentiality of ChiP for chitinolytic growth and indicated that another chitoporin must exist. Indeed, our Tn-seq and mass spectrometry data has revealed *vp0802* as a gene encoding a key outer membrane chitoporin that is structurally distinct from the ChiP chitoporin. This discovery clearly has a biological basis in ChiP-independent chitooligosaccharide transport for efficient growth, and convincingly reveals that phenotypic functional redundancy is a genomic feature that supports overall cell fitness and survival. Notably, the *vp0802* null mutant displayed extremely delayed chitinolytic growth and eventually expressed its native *chiP* gene for assembly of an outer membrane trimeric chitoporin (see Fig 4C). We noted that ChiP levels were scarce in bacterial outer membranes when Vp0802 was abundant which suggests that an underlying regulatory mechanism is in place to produce a uniform chitooligosaccharide import process to the cell. Accordingly, a *vp0802, chiP* double mutant was unable to grow on colloidal chitin, strongly suggesting that these two chitoporins are the main entry points for large chitooligosaccharides into the cell. It is unclear what temporal gene expression mechanisms are at play for *chiP* and *vp0802*. We speculate that these respective chitoporins might be environmental-niche specific or perhaps are induced by different sized chitin polymers. To this end, GlcNAc2 is considered a key inducer of *chiP* expression [13,17,26]. We don't currently know the exact signal or inducer for *vp0802* expression however the colloidal chitin in our experiments likely needed extensive cleavage by secreted bacterial chitinases for GlcNAc derivatives (including GlcNAc2) to reach cell detectable levels. We did note that sole chitinolytic *V. parahaemolyticus* growth may have produced a temporal viable but not culturable (VBNC) condition in our experiments indicated by a decrease in viable plate counts after a logarithmic growth surge (with no change in optical cell density). Rapid depletion of

preferred chitooligosaccharides might have induced a VBNC state, a feature that many *Vibrio* species exhibit upon starvation conditions [54–57]. Moreover, secreted chitinases acting on any remaining larger chitin fragments could provide smaller GlcNAc moieties to support resuscitation.

Vp0802 does not share sequence or structural similarity to the canonical trimeric ChiP chitoporins in *Vibrio* species (S1 Fig), raising the question about its origins and mechanism related to chitooligosaccharide import. Vp0802 appears to be structurally similar to OprD-like single channel porins, including an *Escherichia coli* porin that has been demonstrated to translocate chitohexaose (GlcNAc6) [32] (S1 Fig). *E. coli* is generally considered non-chitinolytic however it appears that environmental chitooligosaccharides can be assimilated when other preferred nutrient resources are scarce. Conserved homologues of Vp0802 (ranging from ~40-100% identity) are broadly distributed in *V. diabolicus, V. cholerae, V. vulnificus, V. anguillarum, V. alginolyticus*, and many other *Vibrio* species (S1 Fig). Notably, specific *Vibrio* strains do not encode a Vp0802 homologue, including the *V. cholerae* El Tor pandemic strains, which appear to utilize ChiP as the dominant chitoporin [13]. Our work thus reveals functional genomic plasticity with respect to chitinolytic growth, with some *Vibrio* species expressing multiple chitoporins. In the case of *V. parahaemolyticus* RIMD2210633, we present evidence that both chitoporins are functional. Future detailed mechanistic experiments will be required to functionally characterize Vp0802 and its homologues, along with how it integrates with ChiP in context of *Vibrio* biology.

OprD family porins are known to exhibit substrate specificity in the translocation of amino acids and other carboxylic acid containing compounds (-COOH) such that the protein family has been renamed Occ to reflect that functionality [34]. For the Vp0802 OprD-like chitoporin discovered here and the *E. coli* chitoporin, there appears to be a dichotomy, as chitooligosaccharides do not contain carboxylic acid groups, instead they contain N-acetyl groups. In the case of *E. coli*, it is well established that chitohexaose can enter its OprD-like porin [32]. Moreover, the small molecule N-acetyl cysteine has been implicated in binding to OprD in *Pseudomonas aeruginosa* [58]. Therefore, evidence for OprD-like porins interacting with at least some types of N-acetyl compounds exists, suggesting that the facilitated diffusion of chitooligosaccharides through these porins is indeed possible and harmonious with our experimental data.

In summary, the use of Tn-seq resulted in the discovery of a critical HexR regulator that functionally impacts on multiple cell phenotypes that are essential for *V. parahaemolyticus* environmental fitness. Additionally, a single channel outer membrane porin (distinct from the canonical trimeric ChiP *Vibrio* chitoporin), was found to be essential for efficient chitinolytic growth. This porin is broadly distributed in many *Vibrio* species, thus revealing a new chitin import pathway which has biological relevance for ocean chitin degradation and carbon mobilization. Our data also highlights HexR dependent *Vibrio* metabolic features that have potential for biotechnological applications that degrade chitinaceous compounds to generate medicinally valuable products such as glucosamine.

## Materials & methods

### Bacterial strains & growth conditions

*Vibrio parahaemolyticus* RIMD2210633 was grown in Luria Miller Broth (LB; Bioshop; LBL417.1), Luria Broth Salt (LBS; 10g tryptone, 5g yeast extract, 20g NaCl, 20mM Tris-HCl pH 8.0), or M9 Minimal Media (MM9; 420mM $Na_2HPO_4$, 220mM $KH_2PO_4$, 86mM NaCl, 187mM $NH_4Cl$, $MgSO_4$, $CaCl_2$, 0.4% w/v glycerol). Cultures were grown at 37°C and 200rpm unless otherwise stated. Antibiotics were used in the growth medium as required:

chloramphenicol (Sigma) at 2.5 $\mu$ g/ml or 30 $\mu$ g/ml (for *V. parahaemolyticus* and *Escherichia coli* respectively), ampicillin at 100 $\mu$ g/ml (for *E. coli),* erythromycin at 10 $\mu$ g/ml (for *V. parahaemolyticus*) and neomycin at 25 $\mu$ g/ml (for *E. coli*). For growth on colloidal chitin, 0.4% colloidal chitin was substituted for glycerol and cultures were grown at 30°C and 250 rpm. For biofilm assays, 0.2% casamino acids (Fluka, 70171) was added to MM9. Agar (Bioshop; AGR003.1) was added to a final concentration of 1.5% (w/v) for solid media preparations unless otherwise stated. Spent Media was prepared by growing *V. parahaemolyticus* in LB, subculturing into LB at a starting $OD_{600nm}$ of 0.025 for 16 hours at 37°C and 200 rpm. Cells were pelleted at 5000rpm and 4°C until supernatant was clear (30 minutes) and then liquid media was filter sterilized using Nalgene Rapid-Flow 0.22 $\mu$ m filter (Thermofisher; 564-0020). Table 1 contains information relating to strains and plasmids used in this study.

## Generation of a streptomycin resistant strain of *V. parahaemolyticus*

Wild type RIMD2210633 was cultured overnight in LBS media without antibiotics overnight at 37°C/200RPM. The culture was sub-cultured into LBS containing 25 µg/mL of streptomycin and incubated at 37°C/200RPM until cell growth was visible, around 48 hrs. Once cell density was visible in the culture tubes, the cells were sub-cultured into LBS containing increasing concentrations of streptomycin until they reliably reached stationary phase after overnight growth at 200 µg/mL streptomycin. This culture was streak-purified and single colonies were screened for their ability to grow in LBS containing 200 µg/mL streptomycin. A colony which reliably reached stationary phase after overnight growth in media containing 200 µg/mL streptomycin was stored and named VpSm.

## Genome sequencing, quality analysis, and mutation analysis

Genome sequencing of the wild type and VpSm strain were completed following phenol-chloroform extraction of genomic DNA from both strains. Following, sequencing libraries were prepared using the Illumina XT DNA Library Preparation Kit (Illumina, FC-131-1024) and sequenced on the Illumina MiSeq platform. Following collection of the raw sequencing data, sequence files were analyzed for quality by FastQC, followed by assembly of the genomes of both strains using SPAdes, with default settings. Following assembly of the genomes, the genome assembly quality was determined using QUAST, with default settings except the minimum contig length for inclusion, which was increased to 700 bp based on the genome assemblies. Finally, Snippy was used to identify mutations present in both strains compared to the *V. parahaemolyticus* RIMD2210633 reference genome, using default settings.

## Transposition experiments

Transposition of the VpSm strain was performed using conjugation between VpSm recipients and SM10λpir *E. coli* pSC189-CmR donors. VpSm and SM10λpir were mixed at a 1:1 ratio normalized by OD600 and washed twice in PBS to remove overnight culture antibiotics. Cells were resuspended in LBS, plated on LBS media on an 0.45µm MLE filter (Millipore), and incubated at 37°C for 4 hours (or various times for the optimization experiments). Cells were collected in 1x MM9 salts by vortexing the filter in a 15 mL conical tube containing 1x MM9 salts, and the cells were then plated on MM9 glucose agar plates containing streptomycin (100 µg/mL) and chloramphenicol (5 µg/mL) to select only VpSm::Himar1 transposon mutants. Cells to be counted for colony-forming units were serially diluted, while transpositions to be used in a selective growth experiment for transposon sequencing were diluted to a total volume of 15 mL in 1x MM9 salts and plated over 150 MM9 glucose plates (FisherBrand, 100mm x 15mm), and incubated at 30°C for 36 hrs prior to collection.

**Table 1. Strains and plasmids used in this study.**

| Strain or Plasmid | Description |
|---|---|
| WT *V. parahaemolyticus* | Wild-type (WT) *V. parahaemolyticus* RIMD 2210633 |
| Δ*hexR* | *V. parahaemolyticus hexR* null mutant |
| Δ*hexR*/pVSV105 | *V. parahaemolyticus hexR* null mutant with pVSV105 |
| Δ*hexR*/pVSV105-*hexR* | *V. parahaemolyticus hexR* null mutant carrying *hexR*-pVSV105, complementation construct |
| WT/pVSV105-*hexR* | *V. parahaemolyticus* overexpressing *hexR* |
| WT/Tn5::*vpa1380* GFP(+) | *V. parahaemolyticus* containing mini-Tn5 transposon, inserted into *vpa1380,* green fluorescence |
| Δ*hexR*/Tn5::*vp0945* GFP(-) | *V. parahaemolyticus hexR* null mutant containing mini-Tn5 transposon insertion into *vp0945*, isolate A |
| Δ*hexR*/Tn5::*vpa1639* GFP(-) | *V. parahaemolyticus hexR* null mutant containing mini-Tn5 transposon insertion into *vp1639*, isolate B |
| Δ*hexR*/Tn5::*vp0424* GFP(-) | *V. parahaemolyticus hexR* null mutant containing mini-Tn5 transposon insertion into *vp0424*, isolate C |
| WT/pP*lafK-gfp* | *V. parahaemolyticus* expressing GFP driven by *lafK* promoter activity, for swarming experiments |
| Δ*hexR*/pP*lafK-gfp* | *V. parahaemolyticus hexR* null mutant expressing GFP driven by *lafK* promoter activity, for swarming experiments |
| Δ*vp0802* | *V. parahaemolyticus vp0802* null mutant |
| Δ*vp0802*/pVSV105 | *V. parahaemolyticus vp0802* null mutant with pVSV105 |
| Δ*vp0802*/pVSV105-*vp0802* | *V. parahaemolyticus vp0802* null mutant carrying *vp0802*-pVSV105, complementation construct |
| Δ*vp0760* (Δ*chiP*) | *V. parahaemolyticus chiP* null mutant |
| Δ*vp0760*/pVSV105 | *V. parahaemolyticus chiP* null mutant with pVSV105 |
| Δ*vp0760*/pVSV105-*vp0760* | *V. parahaemolyticus chiP* null mutant carrying *vp0760*-pVSV105 by conjugation, complementation construct |
| Δ*vp0802*Δ*vp0760* | *V. parahaemolyticus vp0802, vp0760* double null mutant |
| Δ*vp0802*Δ*vp0760*/pVSV105 | *vp0802, vp0760* double null mutant with pVSV105 |
| Δ*vp0802*Δ*vp0760*/ pVSV105-*vp0802* | *vp0802, vp0760* double null mutant with pVSV105-*vp0802*, complementation construct |
| Δ*vp0802*Δ*vp0760*/ pVSV105-*vp0760* | *vp0802, vp0760* double null mutant with pVSV105-*vp0760*, complementation construct |
| WT/pVSVlux-*glgX*/pDSK519 | *V. parahaemolyticus* with pVSVlux-*glgX* and pDSK519 |
| Δ*hexR*/pVSVlux-*glgX*/pDSK519 | Δ*hexR* with pVSVlux-*glgX* and pDSK519 |
| Δ*hexR*/pVSVlux-*glgX*/pDSK519-*hexR* | Δ*hexR* with pVSVlux-*glgX,* expressing *hexR* from pDSK519 |
| WT/pVSVlux-*pgi*/pDSK519 | *V. parahaemolyticus* with pVSVlux-*pgi* and pDSK519 |
| Δ*hexR*/pVSVlux-*pgi* | Δ*hexR* with pVSVlux-*pgi* |
| Δ*hexR*/pVSVlux-*pgi*/pDSK519-*hexR* | Δ*hexR* with pVSVlux-*pgi,* expressing *hexR* from pDSK519 |
| DH5 $\alpha\lambda$ pir | *E. coli* host for oriR6K-dependent plasmid replication |
| pVSV105 | *Vibrio* shuttle vector with multiple cloning site, replication competent in *V. parahaemolyticus* and DH5 $\alpha\lambda$ pir |
| pVSV105-*hexR* | pVSV105 expressing *hexR* from its native promoter |
| pVSV105-*vp0802* | pVSV105 expressing *vp0802* from its native promoter |
| pVSV105-*vp0760(chiP)* | pVSV105 expressing *vp0760* from its native promoter |
| pVSVlux | Promoter-less *luxCDABE* cassette cloned into SmaI site of pVSV105 |
| pEVS104 | Conjugation helper plasmid with mobilization machinery, used in triparental mating |
| pEVS168 | Mini-Tn5 transposon plasmid, containing a promoter-less *gfp*, erythromycin selection, R6K origin of replication |
| pRE112 | Suicide plasmid, R6K origin of replication, for allelic exchange using SacB mediated sucrose selection |
| pRE112- Δ *hexR* | Δ *hexR* allele in pRE112, for allelic exchange |
| pRE112-Δ*vp0802* | Δ*vp0802* allele in pRE112 for allelic exchange |
| pRE112-Δ*vp0760* | Δ*vp0760* allele in pRE112 for allelic exchange |
| pSC189-Cm | Transposition plasmid to generate mutants for Tn-seq |
| pVSVlux-*glgX* | *glgX* promoter cloned upstream of *luxCDABE* cassette in pVSVlux |
| pVSVlux-*pgi* | *pgi* promoter cloned upstream of *luxCDABE* cassette in pVSVlux |
| pP*lafK-gfp* | *lafK* promoter cloned to create a transcriptional reporter, fused to promoter-less GFP gene, in pVSV105 |
| pAT113 | Broad host range plasmid, constitutive GFP expression |
| pDSK519 | Broad host range plasmid, compatible with pVSV105 |
| pDSK519-*hexR* | Complementation construct for *hexR* expression, compatible with pVSVlux |

## Colloidal chitin preparation

Colloidal chitin was prepared using shrimp chitin flakes (Sigma). Briefly, 20g of shrimp chitin flakes were measured and blended into a fine powder using a blender. In a glass beaker, 100mL of concentrated HCl was added slowly to the powdered chitin and stirred gently with a glass rod every 15 min until the mixture was homogenous. The chitin-HCl mixture was added to 2L of distilled H2O, and allowed to precipitate overnight at 4°C. Following, filtration through a buchner funnel and Whatman filter paper was performed using vacuum suction. The moist chitin cake was rinsed with distilled water in the Buchner funnel until the pH was neutral. The chitin cake was aliquoted into 50 mL conical tubes and sterilized by autoclave.

## Generation of insertion sequencing libraries using HTML-PCR

Following transposon mutagenesis and selection in either chitin or glucose containing minimal media, DNA was extracted from the selected population using phenol-chloroform extraction (see Genome Sequencing, Quality Analysis, and Mutation Analysis). HTML-PCR was performed as previously described with a few modifications [59]. Mainly, HTML-PCR does not require blunting of fragmented DNA ends, as previously published, prior to poly-C tailing. 10µg of DNA was fragmented using 182 Fragmentase (NEB) for 30 min followed by DNA purification using AMPure XP Beads for DNA purification (1X volume, Beckman Coulter). Poly-C tailing was performed followed by DNA cleanup as before, and PCR amplification using LG17 and LG18 oligos as previously described. The second PCR was performed using variable primers – depending on the indices required for illumina sequencing (LG503 and LG710 for the glucose sample, along with LG510, LG706 for the chitin sample) – and 1 µL of the product from PCR1 as template. Final PCR amplifications were cleaned up, visualized using agarose gel electrophoresis, and quantified using the Illumina Quant Kit (NEB). Sequencing of these libraries was performed on the Illumina NextSeq platform, using single end reads, 150 bp in length. Table 2 contains a list of primers used in this study.

## TRANSIT analysis of sequencing data

Following sequencing, sequencing output was analyzed for overall quality using FastQC, followed by trimming and genome mapping using TRANSIT's tpp tool against the RIMD2210633 reference genom (NCBI Accession: NC_004603.1 and NC_004605.1), using default settings except the transposon primer window, which was increased to 40 bp from the start of the read due to differences in the HTML-PCR approach. Output from tpp was used to determine success of the transposon sequencing experiments, particularly considering Tn saturation and density of the libraries. TRANSIT was then used to perform two analyses on the data for both chromosomes: a hidden markov model analysis to identify regions of the genome that are essential for growth on either condition, and a zero-inflated negative binomial approach which compared the glucose and chitin condition data following normalization. Both of these analyses were performed using default TRANSIT settings and protein tables generated from gff3 files from the RIMD2210633 reference genome (NCBI Accession: NC_004603.1 and 183 NC_004605.1). Data from these analyses were explored using LibreOffice Calc to sort and identify genes that carried a ZINB adjusted p-value of less than 0.05, and visualized using R and the GViz package, along with modified R scripts. Circos plots were generated from TRANSIT outputs for transposon insertion counts, and HMM essential gene analysis.

**Table 2. Primers used in this study.**

| Primer | Sequence (5' → 3') | Purpose |
|---|---|---|
| LG17 | TCGTCGGCAGCGTCAGATGTGTATAA GAGACAGTTCTAGAGACCGGGGACTTATCAGCC | HTML PCR Transposon primer |
| LG18 | GTCTCGTGGGCTCGGAGATGTGTATA AGAGACAGGGGGGGGGGGGGGGGGG | HTML PCR PolyC primer |
| LG503 | AATGATACGGCGACCACCGAGATCTA CACTATCCTCTTCGTCGGCAGCGTCAGATGTGT | HTML PCR P5 (503) |
| LG510 | AATGATACGGCGACCACCGAGATCTAC ACCGTCTAATTCGTCGGCAGCGTCAGATGTGT | HTML PCR P5 (510) |
| LG710 | CAAGCAGAAGACGGCATACGAGATTCG CCTTAGTCTCGTGGGCTCGGAGATGTGTATAA | HTML PCR P7 (710) |
| LG706 | CAAGCAGAAGACGGCATACGAGATTCGC CTTAGTCTCGTGGGCTCGGAGATGTGTATAA | HTML PCR P7 (706) |
| NT495 | TTGAGCTCAGTACTGGACGAACAACGC | Δ*hexR* construction |
| NT496 | TTGAATTCGCGCTCTGACTTACTGAAATTCTCC | Δ*hexR* construction |
| NT497 | AAGAATTCCGTTATGACAAGCTAAGTCAG | Δ*hexR* construction |
| NT498 | AAGGTACCATGGCGATCACTAACGCTAAGTTGG | Δ*hexR* construction |
| NT499 | AAGCATGCTTAGTACTGACTTAGCTTGTCATAACG | Δ*hexR* complementation |
| NT511 | AAGGTACCGTGTGCTGGCATGTCGTCAT | pVSVlux-*pgi* construction |
| NT512 | AAGGTACCATGTTACAGTAGGTTCCATTCC | pVSVlux-*pgi* construction |
| NT515 | AAGAGCTCTTGACCACCGTCAGCGTTCACG | pVSVlux-*glgX* construction |
| NT516 | AAGGTACCGTCGTGTCATCGGAGATAACTT | pVSVlux-*glgX* construction |
| NT525 | AAGAGCTCCTGCGTTTCAGTGATCAGGATCTCC | Δ*vp0802* construction |
| NT526 | AAGCATGCTTAGTGGAAGCTGTAAGGGATAAG | Δ*vp0802* construction |
| NT527 | AAGAGCTCTGTATTCAAGTCTGCGATGC | Δ*vp0802* construction |
| NT528 | AACTCGAGGCAGTTAACGCTGAAACCTT | Δ*vp0802* construction |
| NT529 | AACTCGAGCCTACGTCTACAAGTTCTTATC | Δ*vp0802* complementation |
| NT530 | AAGGTACCCATAAGCATAAAGGTAAATGC | Δ*vp0802* complementation |
| NT531 | AAGAGCTCAACATCGTGATTAGCGTGTAACG | Δ*chiP* construction |
| NT532 | AACTCGAGAGCAAGTAGGCTTTTCTTTAGGTAAGAC | Δ*chiP* construction |
| NT533 | AACTCGAGTACTTCTAATAAAGAAGCAGC | Δ*chiP* construction |
| NT534 | AAGGTACCTTGGATCGAGTTCGAAGGCTT | Δ*chiP* construction |
| NT535 | AAGGTACCTTAGAAGTAGTATTCAACACC | Δ*chiP* complementation |
| NT540 | ACAGAGCTCCTGAAACCCAACCCTCATGGCTAA | *lafK* promoter amplicon |
| NT541 | AACGGCCGCTTTCAGGCCTTGGGATATTCC | *lafK* promoter amplicon |
| NT542 | AACGGCCGAGTACTGCGATGAGTGGCAGG | *gfp* amplicon, pEVS168 |
| NT543 | AAGCATGCTCAGTTGTACAGTTCATCCATGCC | *gfp* amplicon, pEVS168 |
| NT544 | ACACTCGAGAGTACTGCGATGAGTGGCAGG | Δ*hexR* complementation |

## Recombinant DNA and cloning techniques

DNA fragments of interest were amplified by PCR using primers (Table 2) synthesized by Integrated DNA Technologies (IDT) and *V. parahaemolyticus* genomic DNA as template. All recombinant plasmid DNA constructs were generated using standard techniques with restriction enzymes from New England Biolabs. Plasmid DNA constructs were verified by DNA sequencing.

## Generation of Δ*hexR*, Δ*vp0802*, Δ*chiP* chromosomal mutant strains

Gene deletion fragments for the respective alleles were generated by PCR and cloned into pRE112 following delivery into *V. parahaemolyticus* through tri-parental mating. Chromosomal integrants were selected on LBS containing chloramphenicol and then were subjected to *sacB*-mediated sucrose selection to drive allelic exchange and chromosomal deletions. Mutants with targeted gene deletions were identified by PCR. For the *vp802, chiP* double

mutant, the *vp0802* null strain was used as the parent and was then constructed with a *chiP* gene deletion.

Genetic complementation constructs were generated by PCR amplifying chromosomal DNA from wild type *V. parahaemolyticus* with gene specific primers (Table 2). In each case, the gene specific promoter region (located in the intergenic region upstream from the gene coding sequence) was included in the DNA amplicon that was eventually cloned into plasmid pVSV105 or pDSK519. The resulting plasmids express the respective cloned gene (*hexR*, *vp0802*, or *chiP*) under the control of its native promoter.

### Generation of *lafK* promoter GFP fusion constructs

Primers NT540 and NT541 were used in a PCR reaction to amplify the upstream intergenic region of *lafK* from wild type *V. parahaemolyticus* RIMD2210633 to capture the *lafK* promoter and regulatory elements. Separately, primers NT542 and NT543 were used to amplify *gfp* from pEVS168. The resulting amplicons were subjected to restriction enzyme digests, *lafK* (Sac1/EagI) and *gfp* (EagI/SphI) respectively. Likewise, pVSV105 was digested with SacI and SphI. Following digestions, a triple ligation was performed with the respective digested DNA fragments, prior to transformation into DH5αλpir. The resulting pVSV105-*lafK*-*gfp* plasmid construct was delivered via tri-parental mating with *E. coli* EVS104 into wild type and Δ*hexR* *V. parahaemolyticus.* Transconjugants were selected on LBS supplemented with chloramphenicol and were streak purified.

### Recombinant DNA constructs for luciferase reporter assays

Intergenic regions upstream of *pgi* and *glgX* were PCR amplified. The subsequent amplicons were digested with SacI and KpnI and directionally cloned into digested pVSVlux. The resulting plasmid constructs were delivered via conjugation into wild type and Δ*hexR* *V. parahaemolyticus* strains.

### Transcriptional luciferase reporter assays to assess promoter activity

Wild type, Δ*hexR, and genetically complemented V. parahaemolyticus* strains containing pVSVlux bearing an upstream intergenic region of regulon members (namely *glgX* and *pgi*) were grown overnight in LBS with chloramphenicol. Cells were inoculated into LB at a starting $OD_{600nm}$ of 0.05 at 30°C, 250rpm. Light emission (counts per second (CPS)) and $OD_{600nm}$ measurements were taken in triplicate in a VictorX5 Multi Label Plate Reader.

### Isolation of mini-Tn5 mutant transposants

Briefly, a conjugal tri-parental mating occurred on LBS agar at 28°C and allowed for the delivery of the plasposon pEVS168 [60] into either Δ*hexR* or wild type *V. parahaemolyticus* RIMD 2210633. The mixture was serially diluted and plated on LBS containing 10 $\mu$ g/mL erythromycin and incubated at 22°C for 36 hours to select for transposants. GFP and non-GFP producing strains for both genetic backgrounds were selected for further characterization. Marker retrieval experiments to map transposition insertion sites (disrupted genes) were conducted as previously described [61, 62].

### Preparation of outer membrane fractions from *V. parahaemolyticus* strains

Outer membrane preparations were derived from colloidal chitin grown stationary phase bacterial cultures using a previously described method [26]. Briefly, bacterial cells were sonicated in 50 mM Tris-HCl buffer followed by the addition of N-Lauryl sarcosine to a final

concentration of 0.5% (w/v) which selectively solubilizes the cytoplasmic membrane and maintains outer membrane composition [36]. The mixture was incubated at 25 °C for 30 min. Samples were then centrifuged at 5000 rpm to remove unlysed cells. The supernatant fraction was then centrifuged at 150,000 X $g$ for 30 min. The resulting supernatant and membranous pellet were resuspended in electrophoretic sample buffer for SDS-PAGE analysis to assess protein profiles and efficiency of fractionation. Protein gels were prepared and run as previously described [63] using a Bio-Rad system.

## Sample preparation for mass spectrometry and proteomics

Excised gel bands were cut into 1mm cubes and incubated in destain solution (100mM ammonium bicarbonate, 20% (v/v) acetonitrile) until clear. Destained gel pieces were dehydrated using acetonitrile and subsequently rehydrated using a solution of 1mM tris(2-carboxyethyl)phosphine and 4mM chloroacetamide and incubate for 30-minutes at room temperature (+21C). Excess solution was discarded, and gel pieces were dehydrated using acetonitrile, rehydrated with 100mM ammonium bicarbonate, and dehydrated again with acetonitrile. For digestion, gel pieces were rehydrated with a solution of 100mM ammonium bicarbonate containing 100ng of trypsin (Promega) and incubated at +37C for 16-hours in a Thermomixer (1000rpm). After digestion, peptides were extracted from the gel pieces using two incubations in 1% formic acid, and one in acetonitrile. Each extraction step was carried out in a bath sonicator for 5-minutes. Recovered peptide solutions were concentrated using a SpeedVac and reconstituted in 0.1% (v/v) trifluoroacetic acid. Reconstituted peptides were desalted prior to mass spectrometry analysis using StageTips. Packed StageTips (2-discs, Empore C18) wells were initially conditioned with 100% (v/v) methanol (100 μL) followed by equilibration with 0.1% (v/v) trifluoroacetic acid (100 μL). Loaded peptides were rinsed using 4% (v/v) methanol in 0.1% (v/v) formic acid (200 μL) and eluted using 60% (v/v) methanol in 0.1% (v/v) formic acid (100 μL). All StageTip steps were carried out using a centrifuge with spinning at 250 x g. Eluted peptides were evaporated to dryness in a SpeedVac centrifuge and reconstituted in a solution of 1% dimethylsulfoxide (v/v) in 1% (v/v) formic acid.

Data dependent mass spectrometry analysis was carried out on an Orbitrap Lumos instrument interfaced to an UltiMate 3000 liquid chromatography (LC) system. For analysis, peptide samples were directly separated by an analytical column (75μm x 15 cm 1.8μm bioZen Peptide XB-C18 beads). The gradient ramped from an initial starting condition of 5% mobile phase B (80% acetonitrile, 0.1% formic acid) to 32% B over 39.5-minutes, with sample pickup and column equilibration using an additional 15.5-minutes (45-minute total run time, mobile phase A = 0.1% formic acid in water). Data acquisition on the Orbitrap Lumos utilized a standard data-dependent acquisition scheme. Specifically, the Orbitrap Lumos was globally set to use a positive ion spray voltage of 2200V, an ion transfer tube temperature of 275°C, a default charge state of 2, advanced precursor determination activated, and an RF Lens setting of 45%. MS1 survey scans covered a mass range of 375-1400m/z at a resolution of 60,000 with an automatic gain control (AGC) target of 4e5 (100%) and the injection time set to 'Auto'. Precursors for tandem MS/MS (MS2) analysis were selected using monoisotopic precursor selection ('Peptide' mode), charge state filtering (2-4z), dynamic exclusion (20 ppm low and high, 60s duration, exclude isotopes = TRUE, exclude within cycle = TRUE), and an intensity threshold of 2.5e4. MS2 scans were carried out in the Orbitrap at a resolution of 30,000 using a higher-energy collision dissociation setting of 30%, an AGC target of 2e5 (400%), injection time set to 'Auto', and an isolation window setting of 1.6m/z. The total allowable MS2 cycle time was set to 3s. MS1 data were acquired in profile mode, and MS2 in centroid.

DDA-MS raw data were analyzed using FragPipe software (version 22.0) with MSFragger (version 4.1) using the default workflow settings (PMID: 28394336). Specifically, raw data files were searched using MSFragger against a representative Vibrio parahaemolyticus proteome fasta database (NCBI taxonomy 223926, version 06/2024, 13,642 entries, includes contaminants) using precursor and fragment tolerances of 20 parts-per million (PPM), a fixed modification of 57.02146 at C, and variable modifications of 15.9949 at M and 42.0106 at protein N-termini. Matching spectra were rescored using MSBooster and peptide spectral match (PSM) validation utilized Percolator prior to ProteinProphet scoring to support filtering at <1% FDR at the PSM, peptide, and protein levels. Peptides were quantified using the IonQuant package built into FragPipe with the default settings. Resulting data were parsed to facilitate comparisons and final reporting using R.

## Growth assays

Strains were grown overnight in LBS supplemented with chloramphenicol. Aliquots of cultures were collected, and cells were washed twice with 1x MM9 salts. Strains were normalized to a starting $OD_{600nm}$ of 0.025 in either LB or MM9 (0.4% Glycerol) and inoculated in triplicate into a 96 well plate. $OD_{600nm}$ measurements were taken at 37°C every 20 minutes for 18 hours in a VictorX5 Multi Label Plate Reader. Each experiment was performed with three technical and biological triplicates. For colloidal chitin growth assays, strains were grown overnight in LBS supplemented with chloramphenicol and inoculated at a starting CFU/mL of approximately $10^5$ cells into MM9 (0.4% colloidal chitin). Sampling was done every 6-12 hours post-inoculation and serial dilutions were plated in duplicate on LBS Agar to determine the number of cells present. For some of the colloidal chitin growth experiments, a viable but not culturable (VBNC) state may have been induced at later sampling time points (beyond 48 hours) and after a logarithmic phase of growth and when cells might experience starvation due to rapid consumption of available GlcNAc derivatives. Specifically, we observed a decrease in viable plate counts on LBS media in the absence of a decrease in cell culture density or any appearance of cell aggregates.

## Spent media assay

A wild type *V. parahaemolyticus* strain containing pVSV105-*hexR* was generated via tri-parental mating using helper strain pEVS104 and then subjected to growth in Spent Media along with wild type, *hexR* mutant, and complement strains. Growth assays were performed as previously described with minor adjustments. Flasks of 10mL of each strain were grown for 36 hours at 37°C and 200rpm in Spent Media. $OD_{600}$ measurements were taken at every 6 hours in duplicate.

## Biofilm formation assay

Biofilm assays were performed as previously described [64], with minor adjustments. Biofilms were grown in glass tubes by inoculating cells in LBS containing chloramphenicol overnight followed by subculturing overnight in MM9 at a starting $OD_{600}$ of 0.025. Liquid media was removed, and cells were washed twice in sterile phosphate buffered saline (PBS; 137 mM NaCl, 2.7 mM KCl, 8.1 mM $Na_2HPO_4$, 1.46mM $KH_2PO_4$) before staining in 0.5% crystal violet for 30 minutes. Biofilms were washed twice with PBS to remove excess crystal violet from the tubes. The biofilms were de-stained with 3mL of 95% ethanol. Aliquots were collected and measurements were taken at an absorbance of 565nm in a 1 cm cuvette on a Biophotometer plus. For quantification of biofilms in LB, strains were grown in LBS in glass tubes overnight followed by a subculture into LB at a starting $OD_{600}$ of 0.025. Following consecutive washing with PBS, staining with 0.5% crystal violet, and destaining with 3mL of 95% ethanol, aliquots were diluted 1/10 in 95% ethanol prior to measuring.

## Microscopy

Microscopy was performed as previously described [45,65], with minor adjustments. Overnight cultures were inoculated into LB and MM9 at a starting $OD_{600nm}$ of 0.025 for 16 hours at 37°C, 200rpm. Strains were fixed in 2 $\mu L$ volumes on 1% agarose pads containing 0.5% PBS. A cover slip was added, and slides were immediately imaged using a Zeiss Axio Imager Z1 microscope at 100x via Differential Interference Contrast (DIC) settings. Cell lengths were analyzed and measured using ImageJ Fiji.

For the observation of swarmer cells, swarmer cell differentiation was induced overnight and 5-10mm cubes of agar were cut out to isolate swarmer cell flares and placed cell-side down onto a 1% agarose pad on a glass slide for 30 seconds. A coverslip was added immediately, and slides were imaged at 63X-100X via DIC and fluorescence on a Z2 Axio Imager (Zeiss).

## Motility assays

Swimming motility assays were performed as previously described [66]. Strains were cultured overnight in LBS, normalized to $OD_{600}$ of 0.05 in 1x MM9 salt and stab inoculated into 0.2% (w/v) agar LBS plates. Measurements were recorded after 5 hours of incubation at 37°C. Swimming of each strain was documented via photograph.

## Induction of swarming motility

Swarming agar was prepared as described previously [65]. Briefly, plates containing 40g/L Brain Heart Infusion (EMD) supplemented with 15g/L agar (Bioshop), 4mM $CaCl_2$, and 50mM 2,2'-Bipyridyl (Sigma) were prepared and dried for 10 minutes at room temperature with lids off. The same day, strains were sub-cultured in LB with appropriate antibiotics to an $OD_{600nm} = 0.8$ before being plated in 1μL spots. Plates were sealed with parafilm and incubated overnight at 24°C. Swarming of each strain was documented with an image captured using a scanner.

## Competition assay

Overnight cultures of both GFP and non-GFP producing strains were grown in LBS supplemented with erythromycin at 37°C, 200rpm. The following day, 1 mL of culture was washed twice in 1x MM9 Salts and inoculated into 50 mL MM9 (0.4% Colloidal Chitin) at a starting $OD_{600nm}$ of 0.025. Cultures grew at 30°C, 250rpm. Aliquots of 250uL were collected (at 0 hours, 6 hours, 24 hours, and 48 hours), immediately serially diluted, and plated onto LB. Single colonies were counted (both the total and fluorescent sub-population) to determine the proportion of fluorescent and non-fluorescent strains using Quantity One software (Bio Rad). A Competition Index (CI) was calculated using the equation:

$$CI = (\Delta hexR_f / WT_f) / (\Delta hexR_i / WT_i)$$

Where subscript (f) denotes colony counts (CFU/mL) at final time point (t=48) and subscript (i) denotes colony counts of starting inoculum (t=0). Statistical significance was calculated using un-paired t-test against baseline of CI=1, where a value of 1 indicates no fitness defect.

## Supporting information

**S1 Data. Determination of essential genes in the glucose and chitin conditions and comparison of the two conditions to find conditionally essential genes.**
(XLSX)

**S2 Data. Summary of the statistical analyses of specific genes for fitness defects during chitinolytic growth.**

(XLSX)

**S3 Data. Summary of the statistical analyses comparing specific intra-strain time points during chitinolytic growth.**
(XLSX)

**S4 Data. List of putative HexR regulon gene members based on RegPrecise analysis and score to the consensus sequence derived for the *Vibrio* genus.**
(XLSX)

**S1 Fig. AlphaFold2 structural predictions for (A) Vp0802 of *V. parahaemolyticus*, (B) Overlap between Vp0802 and *E. coli* OprD-like chitoporin (PDB 7vu1), (C) Vp0760 (ChiP) of *V. parahaemolyticus*, (D) Overlap between Vp0760 (ChiP) and VhChiP (PDB 5mdb) of *V. harveyi*, E) Sequence identity for Vp0802 (OprD-like) orthologs in other Vibrio species.** The analysis was performed with Reference Genomes for the indicated strains.
(TIF)

**S2 Fig. SignalP prediction of leader peptide sequences and cleavage sites for Vp0802 and ChiP. Cleavage is predicted at the vertical dotted line.** The leader sequence is located to the left of the dotted line with the initiating methionine as the first amino acid of the unprocessed pre-protein.
(PDF)

**S3 Fig. Genetic complementation analyses during chitinolytic growth.** The indicated mutant strains (magenta) are compared to wild type (WT) *V. parahaemolyticus* harboring plasmid pVSV105 (black). The respective porin gene was cloned under control of its cognate promoter into pVSV105 and mobilized into the indicated bacterial strains.
(TIF)

**S4 Fig. Mass spectrometry data obtained from specific proteins in outer membrane protein preparations.** Samples were derived from wild type bacteria (for Vp0802) (top) and from Δ*vp0802* (for ChiP) (bottom). Yellow highlighted sequences represent peptides accurately identified from the respective tryptic digests by mass spectrometry. In some cases, missed trypsin cleavages within peptides were observed.
(PDF)

**S5 Fig. Genomic region for *hexR* orthologs (yellow) in other *Vibrio* species.** The teal and magenta colours for the *hexR* flanking genes indicate conserved *panP* and *adh* orthologs respectively. Note the presence of a ~300 bp intergenic region between *panP* and *hexR* which likely contains a gene promoter for *hexR* expression. The intergenic region was cloned with the *hexR* ORF (vp1236) and was found to successfully complement a hexR null mutant.
(TIF)

**S6 Fig. Characterization of bacterial strains with transposon insertions that were used for competition experiments.** Top, Schematic showing the genomic location of a mini-Tn5 transposon insertion for the indicated strains. The wild type strain with an insertion in *vpa1380* expresses a transposon-associated GFP allele using a chromosomal promoter, thus producing green fluorescence. The *hexR* mutant transposon insertions are non-fluorescent due to the absence of a suitable promoter position. Bottom, Growth curves in LB for the indicated bacterial strains.
(TIF)

**S7 Fig. Location of nucleotide sequences resembling the HexR consensus binding site.** For the indicated genes, a putative HexR binding site (highlighted yellow) is shown in context of

their respective predicted promoter elements. Promoter elements are shaded in grey with the -35/-10 elements underlined.
(PDF)

## Acknowledgments

Thank you to John Rohde and Maggie Hosmer for the use of plate scanning equipment for the swarming assays.

## Author contributions

**Conceptualization:** Landon J. Getz, Oriana S. Robinson, Nikhil A. Thomas.

**Data curation:** Landon J. Getz, Oriana S. Robinson, Nikhil A. Thomas.

**Formal analysis:** Landon J. Getz, Oriana S. Robinson, Nikhil A. Thomas.

**Funding acquisition:** Nikhil A. Thomas.

**Investigation:** Landon J. Getz, Oriana S. Robinson, Nikhil A. Thomas.

**Methodology:** Landon J. Getz, Oriana S. Robinson, Nikhil A. Thomas.

**Project administration:** Nikhil A. Thomas.

**Resources:** Nikhil A. Thomas.

**Supervision:** Nikhil A. Thomas.

**Validation:** Landon J. Getz, Oriana S. Robinson, Nikhil A. Thomas.

**Visualization:** Landon J. Getz, Nikhil A. Thomas.

**Writing – original draft:** Landon J. Getz, Oriana S. Robinson, Nikhil A. Thomas.

**Writing – review & editing:** Landon J. Getz, Oriana S. Robinson, Nikhil A. Thomas.

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
