## [Decision Letter · Decision Letter 0]

14 Aug 2024

Dear Dr Thomas,

Thank you very much for submitting your Research Article entitled 'Functional genomics of chitin degradation by Vibrio parahaemolyticus reveals finely integrated metabolic contributions to support environmental fitness' to PLOS Genetics.

The manuscript was fully evaluated at the editorial level and by independent peer reviewers. The reviewers appreciated the attention to an important problem, but raised some substantial concerns about the current manuscript. Based on the reviews, we will not be able to accept this version of the manuscript, but we would be willing to review a much-revised version. We cannot, of course, promise publication at that time.

If you decide to revise the manuscript for further consideration at PLOS Genetics, please aim to resubmit within the next 60 days, unless it will take extra time to address the concerns of the reviewers, in which case we would appreciate an expected resubmission date by email to plosgenetics@plos.org.

To resubmit, log into your Editorial Manager account and select the option 'Revise Submission' in the 'Submissions Needing Revision' folder.

We are sorry that we cannot be more positive about your manuscript at this stage. Please do not hesitate to contact us if you have any concerns or questions.

Yours sincerely,

Kai Papenfort

Academic Editor

PLOS Genetics

Sean Crosson

Section Editor

PLOS Genetics

Dear Dr Thomas.

Thank you for submitting your work to PLOS Genetics. The manuscript has now been reviewed by three experts in the field and their comments are provided below. While the referees find your work potentially interesting, they also raised substantial criticisms. We will consider publishing your manuscript only if you can address these criticisms in a revised version of the manuscript.

Kind regards,

Kai Papenfort

Reviewer's Responses to Questions

**Comments to the Authors:**

Reviewer #1: In this work by Getz et al, the authors use a Tn-seq approach to identify genes that play a role in chitin-based growth. The question is timely and relates to an important open question in microbiology. The method is also appropriate and led to two main discoveries. The first discovery is that OprD is a possible novel chitin transporter. The second discovery is that HexR, a TF, is required for growth on chitin but also has more general effects on V. parahaemolyticus physiology. Overall, I found the work interesting and the discoveries quite novel and intriguing. My main concerns are that in many places, the conclusions are overinterpretations of the available results, and parts of the work would benefit from strengthening the supporting data.

Major comments:

1. Lines 172-3 (Fig. 4): If the statistics were performed on the different strains for each timepoint, such a claim (increase in cell numbers between 60 and 80 hrs) should be further supported by statistical analyses between timepoint within the same strain.

2. Line 175 (Fig. 4): Without knowing the statistics of the data for the 24 hour timepoint, the complemented strain appears to have two growth phases (first 24 hours and last 24 hours). Can the author please comment on that? Also, the 60-80 hour phase is the same time frame in which the uncomplemented mutant shows growth, and the statistical analysis in the graph suggests that the difference between the strains is not significant. Can the authors please comment or explain this as well?

3. Line 179 (Fig. 4C): To claim outer membrane localization - did the authors check whether the outer membrane preparations were "pure" or contained the contents of other cellular compartments (e.g. by blotting for known periplasmic or inner membrane components)? Also, another band appears specifically in the mutant at ~25 kDa, and a band between 20-25 kDa is apparent in the WT and complemented strains but not the deletion strain. Can the authors comment on those? Have they tried to identify them?

4. Fig. 5C-D: Can the authors please explain why the growth pattern of the complemented mutant is so different in 5C and 5D? In C, the OD decreases after 10 hours, whereas no decrease in OD is observed in D for up to 25 hours. Unless I misunderstood something, both panels show growth in spent media.

5. Fig. 6B: If I understand the methods correctly, these GFP strains are actually random Tn5 -inserted mutants, and the authors do not know what is the additional mutation. If this is true, then the assay is quite "messy", and the results should be interpreted with caution (or the experiment should be performed on several independent Tn colonies to confirm that any observed phenotype is not due to the random Tn5 mutation).

6. Fig. 7: These results suggest a negative effect of the hexR deletion on swarming. The authors then show a positive effect of the deletion on swimming. Why haven't the authors also quantified swarming mobility on agar to strengthen their claims?

Minor comments:

1. The first paragraph of the Results section (lines 79-85) should include an explanation of the selective chitin conditions. Only selection on glucose is currently mentioned before progressing to descriptions of DNA isolation.

2. Line 100: Can the authors Please explain how are 627 genes essential in both conditions if only 611 were essential for growth on chitin?

3. Lines 160-161: It would be nice to visually show the level of conservation and synteny of these homologs in other vibrios known (or not) to use chitin as a carbon source, to get a sense of the level of conservation of this seemingly important component.

4. Lines 164-6: Can the authors indicate whether the structure prediction was performed on the entire protein sequence containing the signal peptide or after its presumed cleavage? Also, perhaps showing the known or predicted structure of other experimentally investigated OprD family members will provide a better visual appreciation of the similarity of this protein to the rest of the family.

5. Can the authors please add an explanation on how the OprD complementation plasmid prepared?

6. Line 178: Regarding the “Late stationary phase”, can the authors please clarify what timepoint was used to take cells from, since the growth curves are different for each strain.

7. Lines 206-7 (Fig. 5A): I think that all strains reach stationary phase at 400 min, since at this time point the angle of all the curves changes.

8. Lines 227-230 (Fig. 5C): The growth (or rather lack thereof) is similar between the mutant and the wild-type strains. Why are the authors focusing on the mutant here? Also, the decrease in OD600 starting at ~600 min for the complemented strain is interesting. Can the authors comment on whether this can be cell lysis or perhaps aggregation? Have they tried to determine viability by CFU?

9. Lines 237-9 (Fig. 5D): The data points in panel D are few and far apart. It is difficult to make the claims listed here based on the available results. For example, the author can say that the WT growth surpassed the other strains sometime between the 12h and 24h timepoints, but not that it happened "after 24 hours of growth".

10. Fig. 5D: I wonder whether the authors monitored viability (CFU) and not only OD, since OD readings can be misleading, especially after long incubation periods that may lead to the formation of biofilm or cell aggregates that could affect OD readings.

11. Lines 247-8 (Fig. 6A): Since there are no measurements between 4h and 24h, all the authors can say here is that there is a defect during the first 4 hours. However, since the initial OD of the mutant was lower than that of the WT, I am not sure that they can even claim that.

12. Fig. 7A: Please clarify which crystal violet image belongs to which strain.

13. Line 286: Why are the data not shown?

14. Lines 314-6: Can the authors please elaborate on how the predicted promoters were identified?

15. Fig. 8: The conclusions would be strengthened if the authors can show the complementation of HexR downregulates these promoters.

16. Have the authors considered testing whether HexR affects OprD expression directly?

17. Lines 375-7: Have the authors considered testing whether the hexR mutant is indeed deficient in its ability to bind chitin flakes?

18. Line 409: I don’t think that 85% sequence identity is "modest" conservation.

Reviewer #2: Vibrio parahaemolyticus (Vp) is a marine bacterium that can cause gastritis when ingested. As a member of the marine community, like other Vibrios, it plays an important role in nutrient cycling by metabolizing chitinous material, the most abundant polymer in the marine environment. Despite chitin metabolism being well studied in Vibrios for decades, the authors here identify by Tn-seq two new regulators of chitin utilization in Vp. Also shown are effects of mutations in these two genes on growth and phenotypes such as biofilm formation. The manuscript is well written and the rationale described carefully.

Major comments

Lines 201-203. What is the % identity of Vp1236 to Vibrio vulnificus (Vv) HexR? I was unable to find a publication of V. cholerae HexR. An alignment of the Vp1236 and Vv HexR protein sequences would be useful. Where are the amino acid differences? In a putative DNA binding domain (appropriate to show in an alignment)?The same gene name should only be used if the authors demonstrate functional complementation of a Vp hexR deletion mutant with Vv hexR. Or the reverse. An inability to complement could suggest Vp1236 is similar to HexR, but not actually HexR. Are vp1236 and Vv hexR syntenic? This would support the opinion that the two are homologs. A comparison of the organization of the genomic loci for both vp1236 and Vv hexR would also be useful.

What strains are referred to in lines 284-6? And the second half of the sentence is insufficient. What biofilm biosynthetic pathways are being referred to? There are more than one biofilm biosynthesis loci - syp and scv (https://doi.org/10.1128/msystems.01226-21). What “pathways”? There are also multiple signaling pathways that control biofilm formation. Quorum Sensing? c-di-GMP signaling? Also, can't loss of a repressor such as qsvR and/or opaR still lead to biofilm formation despite indicating a mutation in a biofilm pathway? The qualitative term "robust" is not defined. Biofilm results would be bolstered by measuring transcription of the two biofilm biosynthesis loci (syp and scv) by RNA-seq to demonstrate that transcription is unchanged in a hexR deletion mutant. It would also be interesting to explore the coordinate genetic regulation of biofilm formation by this new regulator and those already described. Specifically, one could test whether OpaR and/or QsvR overexpression overrides the biofilm defects seen in the hexR deletion mutant.

Lines 293-301. Designating the filamentous cells as swarmer cells is speculative here. One could measure expression of lafK as in [21] as a criterion of swarmer cell differentiation. Without swarmer cell confirmation, the term “swarmer” should not be used in subsequent lines 298 and 301.

Figure 7. It would be interesting to know whether biofilm formation occurs with chitin as a sole carbon source and what cell morphologies (rod shaped, curved) are observed in the periphery and center of colonies on LB. More details regarding cell lengths would be useful as well.

Minor comments

Line 152. Replace “cell” with “periplasm”

Line 161-2. Citations needed? What are the % AA Similarity and % AA Identity with regard to the other Vibrios?

Lines 166-168. The SignalP analysis is consistent with its export beyond the cytoplasm into the periplasmic space. But provides no information supporting transport or localization to the outer membrane.

Line 171. Since no results are shown that determine an alternative chitin growth pathway, I suggest the authors qualify this statement “…can presumably utilize…”

Line 220. Please provide a more detailed explanation of the complementation here. Plasmid-borne? Approximate copy #? Under control of what promoter?

Line 227. Were the WT cells grown to late stationary phase in LB or MM? With glycerol? With chitin?

Line 292. with chitin? I assume with glycerol based on line 213, but should be clear here too or create an abbreviation for example: "MM9+glyerol" vs "MM9+chitin"

Line 328-329. does location of putative HexR binding site support its role as a DIRECT repressor? For example, is there a putative promoter identifiable? Is the putative binding site near/with the -35/-10?

Fig. 4. Legend has a typo. “b-barrell” should read “�-barrel (beta-barrel)”

Fig 4. What might be the nature of the band at 23-25 kDa that changes in location with the different genetic backgrounds? This was not mentioned.

Fig. 7A. The images shown are of poor quality. I do not know how to interpret these. The images in Panel B could be enlarged and/or an inset used to more clearly show the morphology of individual filamentous and rod-shaped cells. Currently the images are too small to differentiate by eye clearly.

Reviewer #3: This is a clearly written and ecologically relevant paper studying the genes required for growth of V. parahaemolyticus on chitin compared to glucose. The authors use classic Tn-seq experiments to delineate between growth on these two media. They identify expected genes and novel genes, and they chose a few to follow up. My major comment is regarding the organization of the manuscript. There seem to be four important findings (though the authors focus on three): Tn-seq confirms previously known or predicted chitin-growth required genes, T2SS genes, OprD-like porin, and HexR. The authors only briefly describe T2SS and don’t mention it in the abstract or discussion, so it seems like a minor finding in the context of the paper. The majority of the study focuses on HexR. In my opinion, this manuscript could be split into two; keep the current expanded study on HexR, but remove the T2SS and OprD findings, and follow up on OprD in another manuscript. The findings that ChiP is not essential in growth on chitin and that deletion of OprD leads to an increase in ChiP could be another story entirely. Alternatively, the deeper competitive fitness studies of the HexR could be moved to another manuscript and looked at further, leaving this study to broadly focus on the Tn-seq and follow up on three systems: T2SS, OprD, HexR.

Major comments:

• Fig. 4B: I’m not convinced that the data show restoration of growth in the complemented strain. Have the authors looked beyond 80 hrs? The increase in growth is so subtle, and one would expect a complemented strain to grow much more similarly to wild-type. This complementation was on a plasmid, perhaps too high copy (seen in Fig. 4C) is also detrimental to growth? I would suggest a chromosomal complementation.

• Fig. 4B: Are the colonies that eventually grow in the deletion strain suppressors? If re-inoculated, do these now behave as wild-type or closer to it?

• The competitive advantage of the hexR- strain compared to wild-type in mixed cultures is really interesting. I would suggest that the authors test different ratios of mixing to see at what minimum proportion of the total the wild-type can overtake the hexR-. It is difficult to determine the full landscape of competition using only one ratio.

• The finding that deletion of the OprD-like gene leads to an increase in ChiP is really important and seems understudied here. Does deletion of ChiP impact Vp0802? Are there differences in ChiP expression at different stages of growth?

Minor comments:

• Lines 42, 59: Vibrio-mediated

• Line 89: it would help to briefly describe the “chitin condition” here (even though in the methods already) – what is the difference compared to minimal medium with glucose? Just addition of chitin? Is it MM9+0.4% chitin? (line 246)

• Reword sentence starting at 117: “Interestingly, there were several genes not conditionally essential in the chitin condition:” (then list).

• Line 129: Why not include vp1236 in Fig. 2?

• Line 128: Has a previous publication shown that N-acetylated compounds are not transported in OprD porins? This struck me as lack of positive data in other studies, not a known specificity for these proteins. Perhaps reword to eliminate the latter part of the sentence. I also think the reference for the E. coli OprD-like protein that imports chitohexaose would be important to mention in the results section at the same part of the paragraph.

**Have all data underlying the figures and results presented in the manuscript been provided?**

Reviewer #1: Yes

Reviewer #2: Yes

Reviewer #3: Yes

PLOS authors have the option to publish the peer review history of their article (what does this mean?). If published, this will include your full peer review and any attached files.

Reviewer #1: No

Reviewer #2: No

Reviewer #3: No

---

## [Decision Letter · Decision Letter 1]

4 Dec 2024

PGENETICS-D-24-00797R1

Functional genomics of chitin degradation by Vibrio parahaemolyticus reveals finely integrated metabolic contributions to support environmental fitness

PLOS Genetics

Dear Dr. Thomas,

Thank you for submitting your manuscript to PLOS Genetics. After careful consideration, we feel that it has merit but does not fully meet PLOS Genetics's publication criteria as it currently stands. Therefore, we invite you to submit a revised version of the manuscript that addresses the points raised during the review process.

Please submit your revised manuscript within 30 days Jan 03 2025 11:59PM. If you will need more time than this to complete your revisions, please reply to this message or contact the journal office at plosgenetics@plos.org. Please include the following items when submitting your revised manuscript:

We look forward to receiving your revised manuscript.

Kind regards,

Kai Papenfort

Academic Editor

PLOS Genetics

Sean Crosson

Section Editor

PLOS Genetics

Aimée Dudley

Editor-in-Chief

PLOS Genetics

Anne Goriely

Editor-in-Chief

PLOS Genetics

**Additional Editor Comments:**

Dear Dr. Thomas.

Thank you for submitting your revised manuscript. While all referees agreed that the revised manuscript is significantly improved over the initial submission, there are several remaining issues that need to be addressed before publication.

I attached these comments below.

Best wishes

Kai Papenfort

**Journal Requirements:**

Please amend your detailed Financial Disclosure statement. This is published with the article. It must therefore be completed in full sentences and contain the exact wording you wish to be published.

Please ensure that the funders and grant numbers match between the Financial Disclosure field and the Funding Information tab in your submission form. Note that the funders must be provided in the same order in both places as well.

**Reviewers' comments:**

Reviewer's Responses to Questions

**Comments to the Authors:**

Reviewer #1: Getz et al properly addressed my concerns in this revised manuscript. I am atisfied with the additional data and discussion. I am including below a few minor (cosmetic) points for the authors to cosider. Congratulations on an interesting finding!

Minor comments:

Fig. 4B – the X-axis could start at t=0h to show the lag phase decribed in the text.

Lines 181-2: not clear where one parenthesis starts.

Lines 182-4: is something missing in this sentence? It isn’t clearly connecting with the preceding one (maybe the timing in which the described decline is apparent?).

Line 277: is it cell density or CFU?

Fig. 9A: it would be better to spell out the strains rather than to use numbers.

Fig. 9D and lines 364-6: I suggest caution in the interpretation of the swarming results. The differences between the repeats are quite dramatic (diameters range between ~45 and 85 mm for WT, for example). This is bound to affect statistical analysis and confidence.

Reviewer #2: This remains and interesting manuscript describing several genes not implicated prior in metabolism of chitin by Vibrio parahaemolyticus, which plays an important role in global carbon cycling. In this revised manuscript, the authors address most, but not all, comments by this reviewer.

Major comments

1. New lines 216-219. The authors do not adequately address the concern that naming of Vp1236 as HexR is not supported sufficiently. As the authors note, HexR is a member of a family of related proteins HexR/MurR/RpiR. HexR was first named in Pseudomonas aeruginosa in 1990 (https://doi.org/10.1128/jb.172.11.6396-6402.1990). Proctor later reported that P. aeruginosa HexR shared 56% identity with hypothetical regulatory protein in V. cholerae (https://www.proquest.com/dissertations-theses/genetic-biochemical-characterization-hexr/docview/304470780/se-2). This % identity is not reported in the manuscript, only the % identity of among Vibrios. In Pseudomonads, hexR is adjacent to genes zwf and pgl (https://doi.org/10.1111/1751-7915.13263). This is not explained either. The hexR locus of Vibrios contain neither zwf nor pgl, but instead panP and adh, now shown in new S5Fig. So, while vp1236 may encode a transcription factor that participates in controlling some gene similar to those controlled by P. aeruginosa HexR, the evidence presented here and prior do not support that Vp1236 is HexR. Again, functional complementation of a V. parahaemolyticus hexR with P. aeruginosa hexR, for example, would provide rigorous proof. Without such proof, HexR designation is pspeculative.

2. New Fig S6. The GFP containing strains are not isogenic. To be rigorous, strains should carry the GFP marker in the same location. One would also expect the same reduction in growth of the mutant when the marker is swapped: WT is unmarked (GFP-), and the deletion mutant is marked (GFP+).

3. Fig 7B. This panel is qualitative and not interpretable as shown. It should simply be removed.

4. In response to Comment 14 by Reviewer 1, the authors state that “RegPrecise was initially used to identify putative HexR binding sites upstream of specific genes. For the revised manuscript, we selected two genes (pgi and glgX) with HexR binding sites that either scored highly to the consensus HexR binding site or were implicated from other studies.” I recommend a supplemental table listing all the specific genes predicted by RegPrecise to have a putative binding site for Vp1236. Since pgi and glgX “scored highly”, it would be appropriate to show the scores for each gene on the list as well to an explanation of how the genes were prioritized. It is also noted (lines 374-376) that there are no citations to other studies used for selecting pgi and glgX, in contrast to this comment in the rebuttal.

Minor comments

1. New Fig. S7. Please provide the consensus sequence generated from RegPrecise here, as in the text (397-398) and indicate in the yellow highlighted sequence, the nucleotides of the predicted binding sites that match the conserved.

2. New lines 242-248. The authors should acknowledge that by 36 h, the cell density of the uncomplemented deletion mutant exceeded the WT or complemented mutants, with no decline observed. Why might this be?

3. New line 265-267. The above point could be mentioned here as well.

4. The last sentence of the Abstract should read “support” rather than “supports”

Reviewer #3: The authors did an excellent job responding to my comments and those of the other reviewers. It is quite impressive how well they addressed critical method issues and improved the clarity of the manuscript and conclusions. Well done.

**Have all data underlying the figures and results presented in the manuscript been provided?**

Reviewer #1: Yes

Reviewer #2: **No: **As noted in Major comment 4, I feel a table should be included that lists the genes with binding sites for Vp1236, as predicted by RegPrecise.

Reviewer #3: Yes

PLOS authors have the option to publish the peer review history of their article (what does this mean?). If published, this will include your full peer review and any attached files.

Reviewer #1: No

Reviewer #2: **Yes: **Brian Hammer

Reviewer #3: No

**Figure resubmission:**
---

## [Decision Letter · Decision Letter 2]

27 Jan 2025

Dear Dr Thomas,

We are pleased to inform you that your manuscript entitled "Functional genomics of chitin degradation by Vibrio parahaemolyticus reveals finely integrated metabolic contributions to support environmental fitness" has been editorially accepted for publication in PLOS Genetics. Congratulations!

Yours sincerely,

Kai Papenfort

Academic Editor

PLOS Genetics

Sean Crosson

Section Editor

PLOS Genetics

Aimée Dudley

Editor-in-Chief

PLOS Genetics

Anne Goriely

Editor-in-Chief

PLOS Genetics

Comments from the reviewers (if applicable):

Dear Dr Thomas.

Thanks again for submitting your revised manuscript. I am happy to inform you that all referees have now suggested to accept the manuscript and I agree with their opinion. Congrats on a very nice work.

Best wishes

Kai Papenfort

Reviewer's Responses to Questions

**Comments to the Authors:**

Reviewer #2: The authors have addressed my comments and concerns adequately.

**Have all data underlying the figures and results presented in the manuscript been provided?**

Reviewer #2: Yes

PLOS authors have the option to publish the peer review history of their article (what does this mean?). If published, this will include your full peer review and any attached files.

Reviewer #2: **Yes: **Brian Hammer

**Data Deposition**

http://datadryad.org/submit?journalID=pgenetics&manu=PGENETICS-D-24-00797R2

**Press Queries**

---

## [Editor Report · Acceptance letter]

PGENETICS-D-24-00797R2

Functional genomics of chitin degradation by Vibrio parahaemolyticus reveals finely integrated metabolic contributions to support environmental fitness

Dear Dr Thomas,

We are pleased to inform you that your manuscript entitled "Functional genomics of chitin degradation by Vibrio parahaemolyticus reveals finely integrated metabolic contributions to support environmental fitness" has been formally accepted for publication in PLOS Genetics! Your manuscript is now with our production department and you will be notified of the publication date in due course.

With kind regards,

Zsofia Freund

PLOS Genetics

On behalf of:
